# Interpreting CLIP with Hierarchical Sparse Autoencoders

**Vladimir Zaigrajew**[1]  **Hubert Baniecki**[1 2]  **Przemyslaw Biecek**[1 2]

## Abstract

Sparse autoencoders (SAEs) are useful for detecting and steering interpretable features in neural networks, with particular potential for understanding complex multimodal representations. Given their ability to uncover interpretable features, SAEs are particularly valuable for analyzing vision-language models (e.g., CLIP and SigLIP), which are fundamental building blocks in modern large-scale systems yet remain challenging to interpret and control. However, current SAE methods are limited by optimizing both reconstruction quality and sparsity simultaneously, as they rely on either activation suppression or rigid sparsity constraints. To this end, we introduce Matryoshka SAE (MSAE), a new architecture that learns hierarchical representations at multiple granularities simultaneously, enabling a direct optimization of both metrics without compromise. MSAE establishes a state-of-the-art Pareto frontier between reconstruction quality and sparsity for CLIP, achieving 0.99 cosine similarity and less than 0.1 fraction of variance unexplained while maintaining 80% sparsity. Finally, we demonstrate the utility of MSAE as a tool for interpreting and controlling CLIP by extracting over 120 semantic concepts from its representation to perform concept-based similarity search and bias analysis in downstream tasks like CelebA. We make the codebase available at https://github.com/WolodjaZ/MSAE.

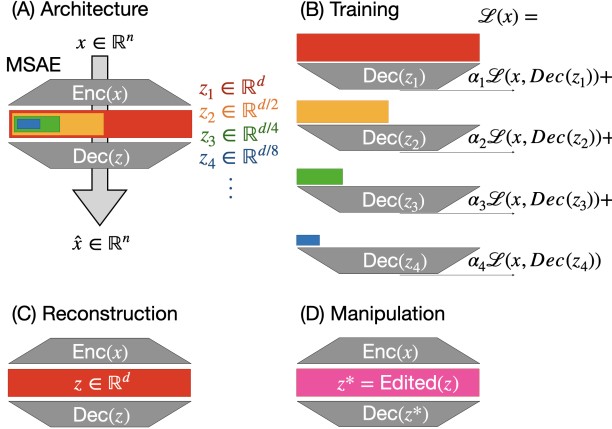

*Figure 1.* 🤗 **Matryoshka Sparse Autoencoder (MSAE)** enables learning hierarchical concept representations from coarse to fine-grained features while avoiding rigid sparsity constraints in TopK and the activation shrinkage problem in ReLU SAE. (B) At training, MSAE uses multiple top-$k$ values up to dimension $d$ instead of a single $k$ like in TopK SAE, combining losses across different granularities. (C) At inference, our method uses the whole $d$-dimensional representation. (D) MSAE allows for more precise editing and manipulation in the concept space.

## 1. Introduction

Vision-language models, particularly contrastive language-image pre-training (CLIP, Radford et al., 2021; Cherti et al., 2023), revolutionize multimodal understanding by learning robust representations that bridge visual and textual information. Through contrastive learning on massive datasets, CLIP and its less adopted successor SigLIP (Zhai et al., 2023) demonstrate remarkable capabilities that extend far beyond their primary objective of cross-modal similarity search. Its representation is a foundational component in text-to-image generation models like Stable Diffusion (Podell et al., 2024) and serves as a powerful feature extractor for numerous downstream vision and language tasks (Shen et al., 2022), establishing CLIP as a crucial building block in modern VLMs (Liu et al., 2023; Wang et al., 2024).

Despite CLIP's widespread adoption, understanding how it processes and represents information remains a challenge. The distributed nature of its learned representations and the complexity of the optimized loss function make it particularly difficult to interpret. Traditional explainability approaches have limited success in addressing this challenge: gradient-based feature attributions (Simonyan, 2013;

[1]Warsaw University of Technology, Warsaw, Poland [2]University of Warsaw, Warsaw, Poland. Correspondence to: Vladimir Zaigrajew <vladimir.zaigrajew.dokt@pw.edu.pl>.

*Proceedings of the 42nd International Conference on Machine Learning*, Vancouver, Canada. PMLR 267, 2025. Copyright 2025 by the author(s).

Shrikumar et al., 2017; Selvaraju et al., 2017; Sundararajan et al., 2017; Abnar & Zuidema, 2020) struggle to provide human-interpretable explanations, perturbation-based approaches (Zeiler & Fergus, 2014; Ribeiro et al., 2016; Lundberg & Lee, 2017; Adebayo et al., 2018; Baniecki et al., 2025) yield inconsistent results, and concept-based methods (Ramaswamy et al., 2023; Oikarinen et al., 2023) are constrained by their reliance on manually curated concept datasets. This interpretability gap hinders our ability to identify and mitigate potential biases or failure modes of CLIP in downstream applications (Biecek & Samek, 2024). Recent advances in mechanical interpretability (Conmy et al., 2023; Bereska & Gavves, 2024) use sparse autoencoders (SAEs) as a tool for disentangling interpretable features in neural networks (Cunningham et al., 2024). When applied to CLIP's representation space, SAEs offer the potential to decompose complex, distributed representations into human-interpretable components through self-supervised learning. It eliminates the need for concept datasets and limits predefined concept sets in favor of natural concept emergence.

However, training effective SAEs poses unique challenges. The richness of data distribution and high dimensionality of CLIP's multimodal embedding space require tuning the sparsity-reconstruction trade-off (Bricken et al., 2023; Gao et al., 2025). Furthermore, evaluating SAE effectiveness extends beyond traditional metrics, requiring the discovery of interpretable features that maintain their semantic meaning across both visual and textual modalities. Current approaches for enforcing sparsity in autoencoders use either $L_1$ (Bricken et al., 2023) or TopK (Gao et al., 2025) proxy functions, each with significant drawbacks. $L_1$ regularization results in activation shrinkage, systematically underestimating feature activations and potentially missing subtle but important concepts. TopK enforces a fixed number of active neurons, imposing rigid constraints that may not align with the natural concept density in different regions of CLIP's embedding space (Gao et al., 2025; Bussmann et al., 2024).

To this end, we propose a hierarchical approach to sparse autoencoders, a new architecture inspired by Matryoshka representation learning (🪆, Kusupati et al., 2022), as illustrated in Figure 1. While Matryoshka SAE (MSAE) can be applied to interpret any neural network representation, we demonstrate its utility in the CLIP's complex multimodal embedding space. At its core, MSAE applies TopK operations $h$-times with progressively increasing numbers of $k$ neurons, learning representations at $h$ granularities simultaneously – from coarse concepts to fine-grained features. By combining reconstruction losses across all granularity levels, MSAE achieves a more flexible and adaptive sparsity pattern. We remove the rigid constraints of simple TopK while avoiding the activation shrinkage problems associated with $L_1$ regularization, resulting in the state-of-the-art Pareto frontier between the reconstruction quality and sparsity.

**Contributions.** We introduce a hierarchical SAE architecture that establishes a new leading Pareto frontier between reconstruction quality (0.99 cosine similarity and $< 0.1$ FVU) and sparsity ($\sim 80\%$), while maintaining computational efficiency comparable to standard SAEs at inference time. We develop a robust methodology for validating discovered concepts in CLIP's multimodal embedding space, successfully identifying and verifying over 120 interpretable concepts across both image and text domains. Through extensive empirical evaluation on CC3M and ImageNet datasets, we demonstrate progressive recovery capabilities and the effectiveness of hierarchical sparsity thresholds compared to existing approaches. We showcase the practical utility of MSAE in two key applications: concept-based similarity search with controllable concept strength and systematic analysis of gender biases in downstream classification models through SAE activations and concept-level interventions on the CelebA dataset.

## 2. Related Work

**Interpreting CLIP models.** CLIP interpretability research follows two main directions: a direct interpretation of CLIP's behavior and using CLIP to explain other models. Direct interpretation studies focus on understanding CLIP's components through feature attributions (Joukovsky et al., 2023; Sammani et al., 2024; Zhao et al., 2024), residual transformations (Balasubramanian et al., 2024), attention heads (Gandelsman et al., 2024), and individual neurons (Goh et al., 2021; Li et al., 2022). Li et al. (2022) discovered CLIP's tendency to focus on image backgrounds through saliency analysis, while Goh et al. (2021) identified CLIP's multimodal neurons responding consistently to concepts across modalities. For model explanation, CLIP is used to analyze challenging examples (Jain et al., 2023), robustness to distribution shifts (Crabbé et al., 2024), and label individual neurons (Oikarinen & Weng, 2023). In this work, we explore both directions in Section 5 via the detection of semantic concepts learned by CLIP using MSAE (Section 5.1) and the analysis of biases in downstream models built on MSAE-explained CLIP embeddings (Section 5.3).

**Mechanistic interpretability.** Mechanistic interpretability seeks to reverse engineer neural networks analogously to decompiling computer programs (Conmy et al., 2023; Bereska & Gavves, 2024). While early approaches focus on generating natural language descriptions of individual neurons (Hernandez et al., 2021; Bills et al., 2023), the polysemantic nature of neural representations makes this challenging. A breakthrough comes with sparse autoencoders (SAEs) (Bricken et al., 2023; Cunningham et al., 2024), which demonstrate the ability to recover monosemantic features. Recent architectural advancements like Gated (Rajamanoharan et al., 2024a) and TopK SAE vari-

ants (Gao et al., 2025) improve the sparsity–reconstruction trade-off, enabling successful application to LLMs (Templeton et al., 2024), diffusion models (Surkov et al., 2024), and medical imaging (Abdulaal et al., 2024). Recent work on SAE-based interpretation of CLIP embeddings (Rao et al., 2024) shows promise in extracting interpretable features.

**Concept-based explainability.** Concept-based explanations provide interpretability by identifying human-coherent concepts within neural networks' latent spaces. While early approaches relied on manually curated concept datasets (Kim et al., 2018; Zhou et al., 2018; Bykov et al., 2023), recent work has explored automated concept extraction (Ghorbani et al., 2019; Kopf et al., 2024) and explicit concept learning (Liu et al., 2020; Koh et al., 2020; Espinosa Zarlenga et al., 2022), with successful applications in out-of-distribution detection (Madeira et al., 2023), image generation (Misino et al., 2022), and medicine (Lucieri et al., 2020). However, existing methods often struggle to scale to modern transformer architectures with hundreds of millions of parameters. Our approach addresses this limitation by first training SAE without supervision on concept learning, then efficiently mapping unit-norm decoder columns to defined vocabulary concepts using cosine similarity with CLIP embeddings.

## 3. 🪆 Matryoshka Sparse Autoencoder

### 3.1. Preliminaries

Sparse autoencoders (SAEs) decompose model activations $x \in \mathbb{R}^n$ into sparse linear combinations of learned directions, aiming for interpretability and monosemanticity. The standard SAE architecture consists of:

$$z = \text{ReLU}\big(W_{\text{enc}}(x - b_{\text{pre}}) + b_{\text{enc}}\big),$$
$$\hat{x} = W_{\text{dec}}z + b_{\text{pre}}, \quad (1)$$

where encoder matrix $W_{\text{enc}} \in \mathbb{R}^{n \times d}$, encoder bias $b_{\text{enc}} \in \mathbb{R}^d$, decoder matrix $W_{\text{dec}} \in \mathbb{R}^{d \times n}$, and preprocessing bias $b_{\text{pre}} \in \mathbb{R}^n$ are the learnable parameters, with $d$ being the dimension of the latent space. The basic reconstruction objective is $\mathcal{L}(x) := \|x - \hat{x}\|_2^2$.

Existing approaches established two primary sparsity mechanisms. ReLU SAE (Bricken et al., 2023) uses $L_1$ regularization with the objective $\mathcal{L}(x) := \|x - \hat{x}\|_2^2 + \lambda\|z\|_1$, while TopK SAE (Gao et al., 2025) enforces fixed sparsity through $z = \text{ReLU}\left(\text{TopK}\left(W_{\text{enc}}(x - b_{\text{pre}}) + b_{\text{enc}}\right)\right)$. However, each approach faces distinct limitations: $L_1$ regularization causes activation shrinkage (Rajamanoharan et al., 2024a), while TopK imposes rigid sparsity constraints. A recent effort to address the rigidity of TopK is BatchTopK (Bussmann et al., 2024), which replaces the standard TopK function with BatchTopK within the TopK SAE method. The BatchTopK function treats all batch activations as a single, flattened vector before applying TopK. This allows for a flexible number of active features per sample, with the total number of active features across the batch averaging to $k \times$ batch size. Although BatchTopK relaxes the fixed sparsity of traditional TopK, it still relies on a predetermined $k$ parameter that requires careful tuning and continues to suffer from the potential for certain features to become 'dead' or rarely activated if they consistently fall outside the TopK selection.

### 3.2. Matryoshka SAE Architecture

Following Matryoshka representation learning (🪆, Kusupati et al., 2022), we propose a SAE architecture that learns representations at multiple granularities simultaneously. Instead of enforcing a single sparsity threshold $k$ or using $L_1$ regularization, our approach applies multiple TopK operations with increasing $k$ values, optimizing across all granularity levels. We set $k$ values as powers of 2, i.e. $k_i = 2^i$ up to dimension $d$, which provides effective coverage of the representation space while maintaining reasonable computational costs. For a given input $x$, MSAE computes $h$ latent representations during training using a sequence of increasing k values $\{k_1, k_2, \ldots, k_h\}$ with $k_1 < k_2 < \ldots < k_h \leq d$:

$$z_i = \text{ReLU}(\text{TopK}_i(W_{\text{enc}}(x - b_{\text{pre}}) + b_{\text{enc}})),$$
$$\hat{x}_i = W_{\text{dec}}z_i + b_{\text{pre}}, \quad (2)$$
$$\mathcal{L}(x) := \sum_{i=1}^{h} \alpha_i \|x - \hat{x}_i\|_2^2,$$

where $\alpha_i$ are weighting coefficients for each granularity level. At inference time, we can either apply TopK with any desired granularity or discard it entirely, leaving only ReLU, which allows the model to utilize all neurons it deems essential for reconstruction.

**Hierarchical learning.** The key insight of our approach is that different samples require different levels of sparsity (numbers of concepts) for an optimal representation. By simultaneously optimizing across multiple $k$ values, MSAE learns a natural hierarchy of features. Our TopK operations maintain a nested structure where features selected at each level form a subset of those selected at higher $k$ values, i.e. $\text{TopK}_1 \subseteq \text{TopK}_2 \subseteq \ldots \subseteq \text{TopK}_h$. Such a hierarchical structure ensures coherence between granularity levels, where low $k$ values capture coarse, high-level concepts while higher $k$ values progressively enable fine-grained feature representation.

**Sparsity coefficient weighting.** We propose and evaluate two strategies for setting the weighting coefficients $\alpha_i$. The *uniform weighting* (UW) approach sets $\alpha_i = 1$ for all $i$, while the *reverse weighting* (RW) strategy uses $\alpha_i = h - i + 1$, giving higher weights to lower $k$ values. By

weighting the loss more heavily for sparser reconstructions, RW encourages the model to learn features that maintain reconstruction quality at lower k values. As shown in Table 1, this results in improved sparsity without significant performance degradation as the model learns that sparse representations achieve better loss even with slightly worse reconstruction quality, compared to UW which focuses primarily on reconstruction quality.

### 3.3. Training and Inference

CLIP embeddings exhibit misalignment across modalities, which can impact SAE training convergence and cross-modal transferability. Following (Bhalla et al., 2024), we normalize embeddings to ensure consistent behavior across modalities. We first center embeddings by subtracting the per-modality mean estimated from the training dataset. Next, we scale the centered embeddings by a dataset-computed scaling factor following (Conerly et al., 2024) to obtain $\mathbb{E}_{x \in \mathcal{X}}[\|x\|_2] = \sqrt{n}$. This scaling ensures that $\lambda$ has consistent effects across different CLIP architectures and modalities. For training, we compute the mean vector and scaling factor from the image modality. During inference on text embeddings, we apply the text-specific mean and scaling factor. Additionally, at inference, we remove TopK constraints from TopK-trained models, allowing the model to adaptively select the number of active features based only on ReLU activation.

## 4. Evaluating MSAE

In this section, we conduct extensive experiments to evaluate MSAE against ReLU and TopK SAEs. We compare the sparsity–fidelity trade-off (Section 4.2), at multiple granularity levels (Section 4.3). We follow with evaluating the semantic quality of learned representations beyond traditional distance metrics (Section 4.4), analyzing decoder orthogonality (Section 4.5), and examining the statistical properties of SAE activation magnitudes (Section 4.6). To verify that MSAE successfully learns hierarchical features, we conduct experiments on the progressive recovery task (Section 4.7). We conclude with an ablation study comparing the influence of different training modalities in Section 4.8. In response to reviewer feedback, we've incorporated an analysis of BatchTopK models within Section 4.2. However, given their performance characteristics were not as competitive, we didn't extend their evaluation to subsequent sections.

**Setup.** All SAE models are trained on the CC3M (Sharma et al., 2018) training set with features (post-pooled) from the CLIP ViT-L/14 or ViT-B/16 model. Image modality is evaluated on ImageNet-1k training set (Russakovsky et al., 2015), while text modality is evaluated on the CC3M validation set. Each SAE is trained with expansion rates of $8\times$, $16\times$ and $32\times$, effectively scaling the latent layer from 768 to

$\{6144, 12288, 24576\}$ neurons for ViT-L/14, and from 512 to $\{4096, 8192, 16384\}$ neurons for ViT-B/16. We provide further details on the implementation and hyperparameter settings in Appendix B.

### 4.1. Evaluation Metrics

Here, we briefly define each metric used to evaluate SAE.

$L_0$ denotes the mean proportion of zero elements in SAE activations.

**Fraction of variance unexplained (FVU)**, also known as Normalized MSE (Gao et al., 2025), measures reconstruction fidelity by normalizing the mean squared reconstruction error $\mathcal{L}(x)$ by the mean squared value of the (mean-centered) input. **Explained variance ratio (EVR)** is FVU's complement metric, defined as $1 - \text{FVU}$.

**Linear probing (LP)** assesses how well SAE preserves semantic information in the reconstructed embeddings on the downstream task. To evaluate this, we train a linear probe model on ImageNet-1k using CLIP embeddings as a backbone, with AdamW optimizer (lr $= 1e-3$), ReduceLROnPlateau scheduler, and batch_size=256. We measure performance by comparing predictions from original versus reconstructed embeddings using two metrics: Kullback-Leibler divergence (KL) between predicted class distributions and classification accuracy (Acc), where accuracy uses $\text{argmax}$ predictions from original embeddings as targets.

**Centered kernel nearest neighbor alignment (CKNNA)** (Huh et al., 2024) measures kernel alignment based on mutual nearest neighbors, providing a quantitative assessment of alignment between SAE activations and input embeddings. A detailed explanation is provided in Appendix E.

**Decoder orthogonality (DO)** calculates the mean cosine similarity of the lower triangular portion of the SAE decoder, where 0 indicates perfect orthogonality. This metric assesses how orthogonal the monosemantic feature directions are in the decoder.

**Number of dead neurons (NDN)** is a metric that measures how many neurons remain consistently inactive (zero in the SAE activations layer) across all inputs during training or evaluation, indicating the network's inability to fully utilize its capacity for learning semantic features.

### 4.2. Sparsity–Fidelity Trade-off

We assess SAE performance using sparsity–fidelity trade-off, measuring sparsity with $L_0$ and reconstruction quality with EVR, following previous work. Figure 2 reveals that ReLU SAE shows difficulty balancing performance, achieving either high fidelity with low $L_0$ or the opposite, with expansion rate primarily improving sparsity. TopK SAE with higher $k$ values achieves better but not ReLU-level

*Table 1.* **Quantitative comparison of SAE models on ImageNet-1k.** We compare the following SAEs with expansion rate 8: ReLU with varying sparsity regularization ($\lambda$), TopK and BatchTopK with 64 or 256 active neurons, and Matryoshka using uniform (UW) or reverse weighting (RW) $\alpha$ coefficients. Arrows ($\uparrow$/$\downarrow$) indicate the preferred direction of metrics. NDN values in parentheses show the dead neuron count on the training set. LP (KL) values are scaled by $10^6$ for readability. Extended results for higher expansion rates and the text modality are reported in Appendix F.5.

| Model | $L_0 \uparrow$ | FVU $\downarrow$ | CS $\uparrow$ | LP (KL) $\downarrow$ | LP (Acc) $\uparrow$ | CKNNA $\uparrow$ | DO $\downarrow$ | NDN $\downarrow$ |
|---|---|---|---|---|---|---|---|---|
| ReLU ($\lambda = 0.03$) | $.920_{\pm.008}$ | $.185_{\pm.031}$ | $.928_{\pm.009}$ | $50.5_{\pm77.1}$ | $.977_{\pm.149}$ | $.742_{\pm.005}$ | $.002$ | $0(0)$ |
| ReLU ($\lambda = 0.003$) | $.649_{\pm.007}$ | $.004_{\pm.000}$ | $.998_{\pm.000}$ | $0.66_{\pm1.03}$ | $.994_{\pm.083}$ | $.781_{\pm.004}$ | $.003$ | $0(0)$ |
| TopK ($k = 64$) | $.950_{\pm.009}$ | $.172_{\pm.026}$ | $.912_{\pm.013}$ | $60.1_{\pm90.8}$ | $.930_{\pm.255}$ | $.762_{\pm.004}$ | $.002$ | $0(335)$ |
| TopK ($k = 256$) | $.900_{\pm.004}$ | $.011_{\pm.003}$ | $.994_{\pm.002}$ | $2.71_{\pm5.40}$ | $.987_{\pm.114}$ | $.874_{\pm.003}$ | $.003$ | $0(296)$ |
| BatchTopK ($k = 64$) | $.877_{\pm.012}$ | $.162_{\pm.022}$ | $.917_{\pm.011}$ | $56.9_{\pm85.8}$ | $.931_{\pm.253}$ | $.769_{\pm.004}$ | $.002$ | $0(1477)$ |
| BatchTopK ($k = 256$) | $.882_{\pm.005}$ | $.010_{\pm.005}$ | $.995_{\pm.002}$ | $2.42_{\pm5.12}$ | $.988_{\pm.108}$ | $.860_{\pm.003}$ | $.002$ | $3(919)$ |
| Matryoshka (RW) | $.829_{\pm.008}$ | $.007_{\pm.003}$ | $.997_{\pm.002}$ | $3.13_{\pm7.08}$ | $.987_{\pm.115}$ | $.809_{\pm.002}$ | $.002$ | $2(4)$ |
| Matryoshka (UW) | $.748_{\pm.006}$ | $.002_{\pm.001}$ | $.999_{\pm.000}$ | $0.35_{\pm0.82}$ | $.995_{\pm.070}$ | $.848_{\pm.003}$ | $.002$ | $0(22)$ |

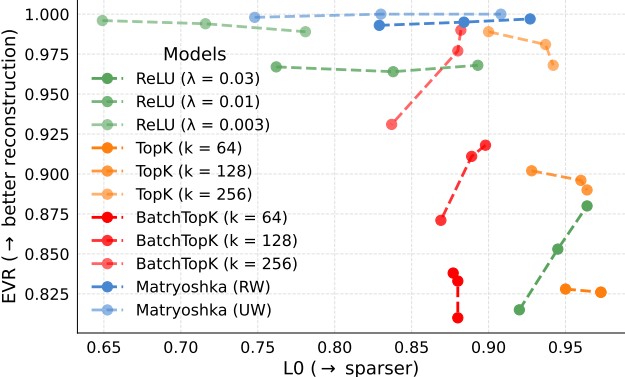

*Figure 2.* **Comparison of sparsity–fidelity trade-offs across SAE architectures on ImageNet-1k.** Each model presents results from all 3 expansion rates, comparing ReLU SAE ($\lambda = \{0.03, 0.01, 0.003\}$), TopK SAE ($k = 64, 128, 256$), BatchTopK SAE ($k = 64, 128, 256$) and MSAE (RW, UW). The optimal SAE would occupy the upper right corner, achieving both high sparsity and reconstruction fidelity. For extended results across both modalities, refer to Figure 11.

As an ablation, we evaluate Cosine Similarity as an alternative reconstruction metric, motivated by observations that SAEs primarily struggle with embedding magnitude reconstruction and CLIP embeddings are commonly $L_2$-normalized. Results in Appendix F.1 show consistent findings, with MSAE showing even clearer advantages through stable, low-variance performance across both modalities.

### 4.3. Ablation: Matryoshka at Lower Granularity Levels

We train both MSAE variants (RW and UW) on two granularities $[128, 256]$ and compare them against TopK with $k = 128$ and $k = 256$ to analyze MSAE behavior at lower granularity levels. Figure 3 shows that Matryoshka achieves similar sparsity to at least the lower TopK variant while maintaining CKNNA and EVR performance comparable to the best TopK variant, and even better with MSAE RW. This demonstrates that even at small granularity, MSAE maintains or improves the Pareto frontier over TopK across various metrics, with RW achieving better trade-offs. As observed also in Section 4.2, MSAE's performance advantages over TopK increase at higher expansion rates.

### 4.4. Semantic Preservation Analysis

In Section 4.2, we only evaluated SAEs using $L_0$ for activation sparsity and EVR for reconstruction fidelity, however these metrics have limitations. $L_0$ only counts active neurons without assessing how well SAE representations align with original embeddings, and EVR focuses solely on distance reconstruction rather than semantic preservation. To address these limitations, we introduce additional metrics. Following (Yu et al., 2025), we adopt the CKNNA metric to assess how well SAE activations preserve the neighborhood structure of CLIP embeddings. We also evaluate semantic preservation through linear probing metrics (Gao et al., 2025; Lieberum et al., 2024), we use LP (KL) to measure prediction distribution alignment and LP (Acc) to compare

fidelity while offering improved sparsity yet consistently suffering from at least 5% of dead neurons (Table 1). While the BatchTopK variant performs similarly to TopK, it exhibits higher fidelity with lesser sparsity when trained on the same $k$. Both variants of MSAE achieve better sparsity than ReLU and better fidelity than TopK or BatchTopK, establishing a superior Pareto frontier while maintaining less than 1% of dead neurons. The RW variant further improves sparsity as expected, with only minor fidelity degradation. Notably, only Matryoshka consistently improves on both metrics with higher expansion rates, while TopK struggles with reconstruction, BatchTopK with sparsity, and ReLU shows improvements only in the highest $\lambda$ at increased expansion rates.

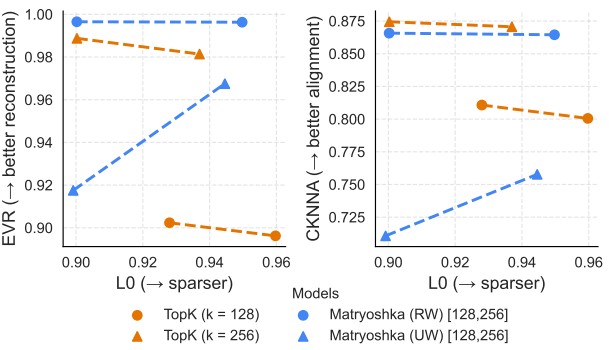

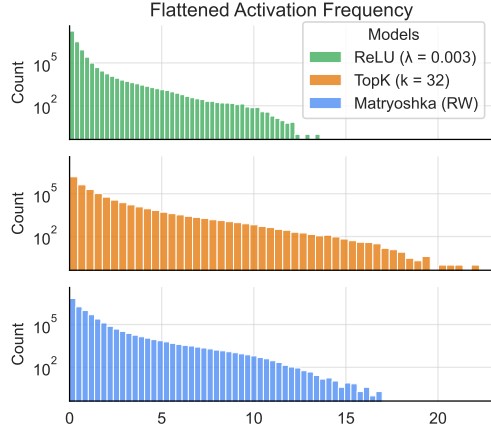

*Figure 3.* **Low Granularity Level Matroshka vs. TopK SAE on ImageNet-1k.** We report FVU (left) and CKNNA (right) metrics for two TopK variants ($k = 128, 256$), and Matryoshka trained on these granularities in RW and UW variants at expansion rates 8 and 16. Even at this small granularity, MSAE improves the Pareto frontier relative to both TopK variants, pushing it as the expansion rate grows from 8 to 16. For extended results across other metrics, refer to Figure 13.

classification accuracy. All metrics are defined in Section 4.1 and presented in Table 1. Our analysis reveals that while cosine similarity and FVU correlate well with linear probing metrics, the alignment metric demonstrates Matryoshka's strength in preserving semantic structures.

### 4.5. Orthogonality of SAE Features

SAEs can disentangle polysemantic representations into monosemantic features, as shown and explained by (Bricken et al., 2023). To evaluate feature monosemanticity, we measure decoder orthogonality using the DO metric, with results reported in Table 1. While all methods achieve high orthogonality as indicated by low DO values, none reach perfect orthogonality. This might stem from multiple factors, including feature absorption as noted in (Chanin et al., 2025), or just learning similar concepts (such as different numbers). We argue that understanding these sources of non-orthogonality is crucial for advancing the development of more effective monosemantic feature learning in SAEs.

### 4.6. Activations Magnitudes Analysis

To analyze the impact of sparsity proxies on SAE, we examine non-zero activation distributions across ViT-L with expansion rate 8 in Figure 4. Matryoshka models display a distinctive double-curvature distribution similar to ReLU-based models, with values between 5 to 10 appearing almost linear in $\log_{10}$ space. Following (Templeton et al., 2024), we attribute low activations to reconstruction purposes rather than semantic meaning. The second curvature reflects natural images' complexity, which requires multiple concept reconstructions rather than single dominant features, as evidenced by the small number of very high values corre-

*Figure 4.* **Distribution of non-zero SAE activations on ImageNet-1k validation set.** Frequency histograms for ReLU ($\lambda = 0.003$), TopK ($k = 32$), and Matryoshka (RW) models at expansion rate 8. Matryoshka models exhibit a double-curvature distribution similar to ReLU models but without activation shrinkage, while TopK shows this pattern only at higher $k$ values, as can be seen in an extended Figure 14. Extended results for higher expansion rates are reported in Figure 15.

sponding to rare, nearly singular concept images (Figure 10). As the sparsity parameter $k$ in TopK methods increases (Figure 14), the transition from one to double-curvature behavior suggests that stronger sparsity constraints create composite features, supported by Appendix C showing that high-activation features ($> 15$) in TopK methods have a lower ratio of valid named features compared to Matryoshka.

### 4.7. Progressive Recovery

To verify that our method learns hierarchical structure, we perform a progressive reconstruction task by using an increasing number of SAE activations, ordered by magnitude, to recover the original vector. Figure 5 shows that reconstruction quality improves with decreasing sparsity thresholds (increasing $k$) during inference. TopK variants exhibit performance plateaus shortly after their training thresholds ($k = \{32, 64\}$), while ReLU-based models show continued improvement but with inferior performance at higher sparsity. MSAE demonstrates a better hierarchical structure that combines TopK's efficient high-sparsity performance with ReLU's scaling capabilities. While our method performs slightly below TopK ($k = 32$) at the highest sparsity, it quickly surpasses TopK's plateau at lower sparsity, achieving performance levels above ReLU models. We observe similar patterns in the CKNNA alignment metric, with MSAE outperforming both TopK ($k = 32$) and ReLU models beyond $k = 10$ while performing only slightly below TopK ($k = 256$) at the lowest sparsity. Evidence of improved hierarchical feature learning across metrics and modalities is presented in Appendix F.4.

*Table 2.* **Training modality influence on MSAE performance.** We train MSAE on the text version of the CC3M train set and compare it to models trained on its original image version, evaluating across both domains using the CC3M validation text set and ImageNet-1k. While models perform best on their training modality, text-trained variants show better cross-domain generalization. **Bold** values indicate the best performance per metric, with NDN showing dead neuron count from the final checkpoint.

| Matryoshka SAE variant | Language metrics on CC3M | | | | Vision metrics on ImageNet-1k | | | | NDN ↓ |
|---|---|---|---|---|---|---|---|---|---|
| | $L_0$ ↑ | FVU ↓ | CS ↑ | CKNNA ↑ | $L_0$ ↑ | FVU ↓ | CS ↑ | CKNNA ↑ | |
| Image (RW) | $.824_{\pm.029}$ | $.060_{\pm.052}$ | $.971_{\pm.026}$ | $.775_{\pm.001}$ | $.829_{\pm.008}$ | $.007_{\pm.003}$ | $.997_{\pm.002}$ | $.809_{\pm.002}$ | 4 |
| Image (UW) | $.755_{\pm.024}$ | $.026_{\pm.027}$ | $.988_{\pm.012}$ | $\mathbf{.790}_{\pm.002}$ | $.748_{\pm.006}$ | $\mathbf{.002}_{\pm.001}$ | $\mathbf{.999}_{\pm.000}$ | $.848_{\pm.003}$ | 22 |
| Text (RW) | $\mathbf{.841}_{\pm.014}$ | $.008_{\pm.003}$ | $.996_{\pm.002}$ | $.782_{\pm.008}$ | $\mathbf{.841}_{\pm.014}$ | $.008_{\pm.003}$ | $.996_{\pm.002}$ | $.782_{\pm.008}$ | 0 |
| Text (UW) | $.791_{\pm.010}$ | $\mathbf{.001}_{\pm.001}$ | $\mathbf{.999}_{\pm.000}$ | $.784_{\pm.007}$ | $.799_{\pm.012}$ | $.015_{\pm.013}$ | $.993_{\pm.006}$ | $\mathbf{.877}_{\pm.003}$ | 0 |

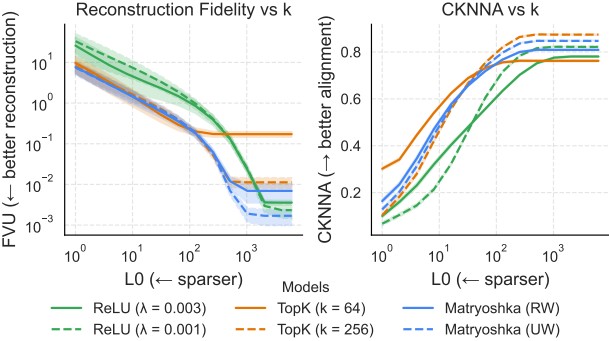

*Figure 5.* **Progressive recovery performance on ImageNet-1k.** We report FVU (left) and CKNNA (right) metrics for different SAE architectures with expansion rate 8 as functions of increasing top $k$ values by magnitudes of SAE activations during inference. SAE trained with TopK variants ($k = 32, 64$) show performance plateaus beyond their training thresholds, while ReLU-based models ($\lambda = 0.001, 0.003$) and Matryoshka variants (UW and RW) demonstrate continuous improvement. Extended results for higher expansion rates and across other metrics are reported in Figures 16 & 17.

## 4.8. Training Modality: Language and Vision

We evaluate how training modality affects MSAE performance by comparing models trained on text with the original image-trained models, validating both modality models across text and image domains in Table 2. While both variants perform best in their training domains, text-trained models achieve superior cross-modal performance, demonstrating stronger generalization capabilities. Moreover, text-trained models achieve higher sparsity on both modalities with no dead neurons, showing better utilization of learned features. These findings position text training as a preferred approach for multi-modal applications where balanced performance is desired. Future research could explore training SAEs on varying ratios of text and image data to optimize cross-modal performance or try to train *crosscoders* (Lindsey et al., 2024) on both modalities simultaneously. We defer extended MSAE evaluations, to Appendix F.

## 5. Interpreting CLIP with MSAE

In this section, we demonstrate how MSAE can enhance interpretability and control interpretable features in CLIP-based applications. We first establish neuron-concept mappings in the activation layer through an automated technique described in Section 5.1. Then, we show its effectiveness in concept-based similarity search across the ImageNet validation set, enabling retrieval of images with varying degrees of explicit concept presence. Moreover, we leverage MSAE to study potential conceptual biases in a gender classification model trained on the CelebA dataset (Liu et al., 2015).

### 5.1. Concept Naming

While self-supervised training of SAE enables learning up to $d$ monosemantic concepts, mapping these concepts to specific neurons remains non-trivial. Previous work used LLMs for identifying neuron-encoded concepts (Bills et al., 2023), but we adopt the more efficient method for CLIP-trained SAE proposed in (Rao et al., 2024), which leverages CLIP's representation space. Our concept detection and validation methodology is detailed in Appendix A, with comprehensive results on valid concept counts across SAE models presented in Table 3. Figure 8 demonstrates the highest-activating text and image examples for the best-matched feature concept 'face' across ReLU, TopK, and MSAE. The consistent concept presence across diverse inputs observed primarily in MSAE variants suggests that only Matryoshka-based methods were capable of learning this monosemantic feature. Supplementary analysis of highly activated concept examples in Appendix G.1 showcases SAE's ability to learn a wide range of concepts, from simple textures and colors to more complex ones like *light* (lights in darkness), countable concepts like *trio* (groups of three), and even nationality-related concepts like *ireland* or *germany*.

Naming SAE features enables using SAE to conduct diverse interpretability analyses related to CLIP. We present two use cases where we apply the MSAE RW variant with an expansion rate of 8.

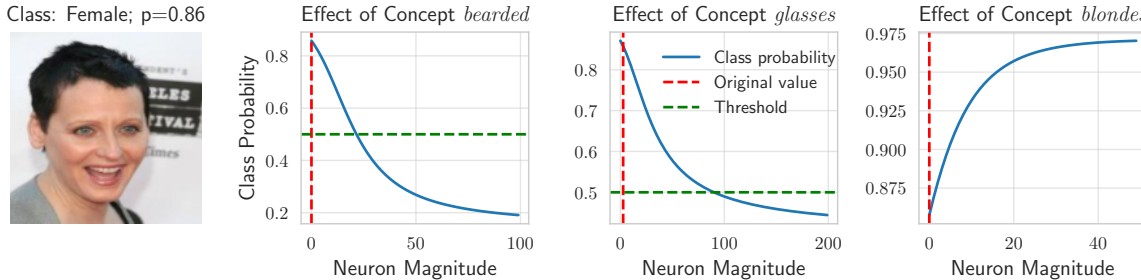

Figure 6. **Impact of concept manipulation on gender classification.** By increasing concept magnitudes (*bearded*, *glasses*, *blonde*) in SAE space and mapping back to CLIP space, we observe changes in gender classification probabilities. Results reveal the model's learned gender associations through plateauing effects: *bearded* and *glasses* bias toward male classification, while *blonde* bias toward female.

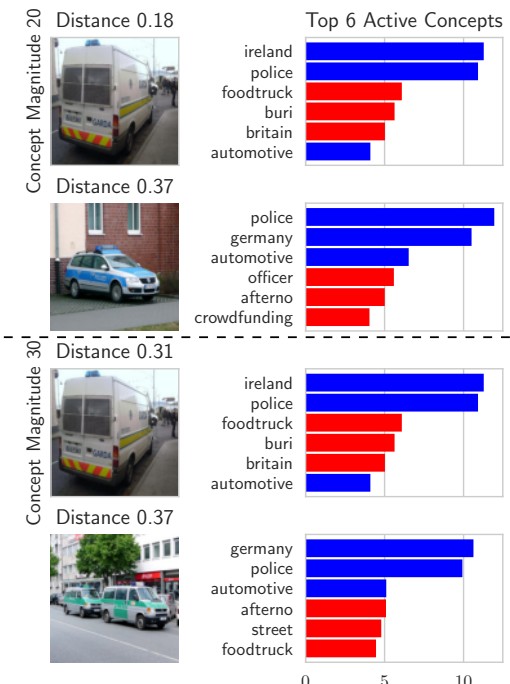

Figure 7. **Nearest neighbor analysis with enhanced *germany* concept.** By increasing the magnitude of the *germany* concept in SAE space (from 0.3 to 20, then 30) and mapping back to CLIP space, we observe shifts in nearest neighbors. While the input image remains the top match (with increasing distance), the second-nearest neighbor changes from a British police vehicle (shown in Figure 21) to a German one.

## 5.2. Similarity Search

CLIP embeddings are widely used for cross-modal similarity search between images and text through cosine similarity metric, primarily for retrieval engines. We extend this capability using SAE in three ways.

First, SAE provides interpretable insights into nearest neighbor (NN) image retrievals. Figure 21 shows the top 8 concepts for the two closest retrieved images, revealing shared semantic concept patterns and explaining why both NNs match the query image of an Irish police vehicle, with the first NN (Irish police vehicle) being closer than the second (British police vehicle). Second, we compare similarity search in CLIP embedding space against SAE activation space using Manhattan distance (detailed and visualized in Appendix G.2). While the first NN remains consistent across both spaces, the second NN in the SAE space shows the same vehicle type from a different angle, demonstrating that similarity searches can be done in both spaces while SAE enables additional concept-based interpretability. Finally, we demonstrate a controlled similarity search by manipulating concept magnitudes. In Figure 7, increasing the *germany* concept strength preserves the original image as the top match but shifts the second NN from an Irish to a German police vehicle, while preserving the overall input image structure. The increasing distances from the original image embedding show how larger magnitude adjustments affect embedding coherence.

## 5.3. Bias Validation on a Downstream Task

CLIP models are commonly used as feature extractors for downstream tasks, enabling efficient fine-tuning with limited data. With MSAE, we can investigate whether downstream models learn to associate specific concepts with classes. To demonstrate this, we train a single-layer classifier on CLIP embeddings from the CelebA dataset to perform binary gender classification (1 for female, 0 for male), achieving an F1 score of approximately 0.99. Through statistical analysis in Appendix G.3, we uncover several concept-gender associations: *bearded* biases toward male classification, *blonde* toward female, and *glasses* showing modest male bias. To validate these findings, Figure 6 demonstrates an example of how increasing these concepts' magnitudes affects classification scores for a female example (see Figure 23 for a male example). The results confirm our statistical analysis, and the plateaus in classification probabilities as concept magnitudes increase help quantify the strength of concept-gender associations in the model.

| ReLU | TopK | MSAE |
|---|---|---|
| **Invalid** feature 5367 with similarity 0.20 | **Invalid** feature 0977 with similarity 0.60 | **Valid** feature 4289 with similarity 0.47 |

**more sparse**

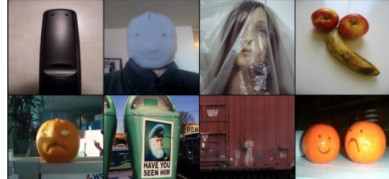 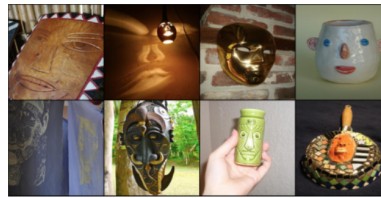 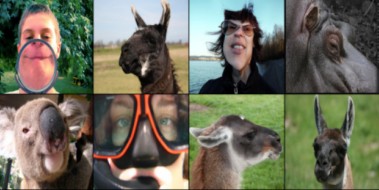

1. how we wish these faces were not covered with huge beards !
2. a row of trees welcomes guests to grant 's .
3. person with a lovely smile
4. person and film character by man *
5. person looks like a cage !
6. you are looking at person .
7. video game series : an ugly figure of a soldier and his mustache
8. person -- looks like a cat !

1. head of a woman 1963.679
2. a face drawn in a tree trunk
3. close - up of a human head consisting with heavy rains depicting headache
4. abstract digital art image of a woman 's face close up on a colorful textured background .
5. face of a young woman
6. face of a female mannequin close - up
7. this face mask looks a lot like film character
8. close - up face carved into a pumpkin .

1. face of a female mannequin close - up
2. face of a young woman
3. view of an intense woman looking into the camera
4. head of a woman 1963.679
5. portrait of a girl with sunlight on her face
6. close - up of the head and beak of animal
7. close up of a wild bull showing details of its face
8. close - up of a young man looking sideways

**better reconstruction**

| **Invalid** feature 5687 with similarity 0.34 | **Invalid** feature 1098 with similarity 0.28 | **Valid** feature 1098 with similarity 0.43 |
|---|---|---|

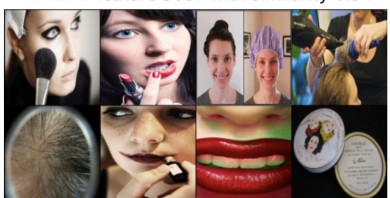 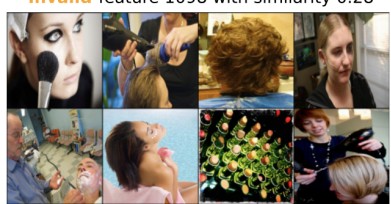 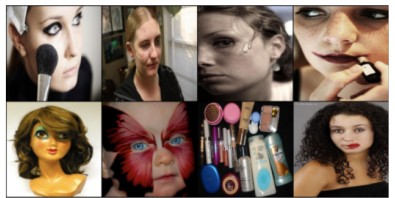

1. medium hairstyles for round faces are pretty fun and interesting if you know a thing about them .
2. wedding hairstyles with garment over the face
3. look 2 , buff a bright color underneath the eye and into inner rim of waterline .
4. highlight the eyelashes in beauty salon .
5. beauty and makeup concept with half face of a beautiful young woman .
6. bridal makeup - final touches before the wedding .
7. steps for a fresh face !
8. how to draw on eyebrows with a pencil

1. look 2 , buff a bright color underneath the eye and into inner rim of waterline .
2. beauty and makeup concept with half face of a beautiful young woman .
3. actor , eye if i wore makeup , this would be perfect .
4. bridal makeup - final touches before the wedding .
5. highlight the eyelashes in beauty salon .
6. sometimes with eye makeup , less is best .
7. face of a female mannequin close - up
8. todays 1950s hair & make up inspiration from actor

1. look 2 , buff a bright color underneath the eye and into inner rim of waterline .
2. face of a female mannequin close - up
3. beauty and makeup concept with half face of a beautiful young woman .
4. actor , eye if i wore makeup , this would be perfect .
5. sometimes with eye makeup , less is best .
6. portrait of a girl with sunlight on her face
7. face of a young woman
8. how to draw on eyebrows with a pencil

*Figure 8.* **Comparison of top-activating examples for the 'face' concept across SAE methods.** Through an automated interpretability process, we identified the best matching SAE neuron for the *face* concept, quantified by a similarity score and a validation status (Valid or Invalid). For this neuron, we then identified the highest activated examples in both text and image modalities across TopK, ReLU, and MSAE, each presented in variants optimized for either sparsity or reconstruction. Both text and image examples with the highest activation values strongly confirm. confirm the concept's presence and demonstrate that only MSAE variants learned the concept of the *face*. We show additional examples of valid concepts with their top-activating examples in Figure 19.

# 6. Conclusion

We propose Matryoshka SAE to advance our understanding of CLIP embeddings through hierarchical sparse autoencoders. MSAE improves upon both TopK and ReLU approaches, achieving superior sparsity–fidelity trade-off while providing flexible sparsity control via the $\alpha$ coefficient. Our experiments demonstrate MSAE's effectiveness through near-optimal metrics, progressive feature recovery, and extraction of over 120 validated concepts, enabling new applications in concept-based similarity search and bias detection in downstream tasks.

**Limitations and future work.** MSAE faces three limitations with clear paths for future improvement. The current implementation's use of multiple decoder passes with different TopK activations introduces computational overhead, which could be addressed through optimized CUDA kernels enabling parallel processing of multiple granularities. While

we demonstrated MSAE's effectiveness using CLIP embeddings, it has great potential to explain hierarchical representations in other embedding spaces, such as SigLIP (Zhai et al., 2023) or modality-specific representations. Finally, since not all neurons correspond to simple concepts in our vocabulary, investigating complex semantic features through LLM-based interpretability methods could provide deeper insights into the learned hierarchical representations. In concurrent independent work, Bussmann et al. (2025) propose an MSAE approach to interpreting language models.

# Impact Statement

This paper presents work whose goal is to advance the field of Machine Learning Interpretability. There are some potential societal consequences of our work, none which we feel must be specifically highlighted here.

## Acknowledgement

Work on this project was financially supported from the SONATA BIS grant 2019/34/E/ST6/00052 funded by Polish National Science Centre (NCN). We also thank the anonymous reviewers for their useful comments.

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

# Appendix for "Interpreting CLIP with Hierarchical Sparse Autoencoders"

# A. Concept Discovery and Validation

Here, we describe our approach for detection, which concepts SAE learned, and how we validated the mappings of these concepts to specific neurons. While LLMs are commonly used to identify neuron-encoded concepts (Bereska & Gavves, 2024; Conmy et al., 2023), we follow Rao et al. (2024) in implementing a more computationally efficient approach, which is more tailored to the CLIP-based SAE.

**CLIP-Based concept matching.** The method uses predefined vocabulary of concepts (e.g., 'hair', 'pink') to compute cosine similarity between CLIP embeddings and SAE decoder columns. After mapping concepts to CLIP's embedding space and applying the same preprocessing as during SAE training, we remove $b_{pre}$ from the preprocessed CLIP embeddings for comparison with the decoder. For the feature columns in the SAE decoder, which are unit-magnitude by definition, the best matching concept to the neuron is determined by maximizing cosine similarity, where a value of 1 indicates perfect alignment. Thus, the optimal concept $s_c$ for neuron $p_c$ is defined as:

$$s_c = \arg\max_{v \in V} \left[\cos(p_c, \text{CLIP}(v))\right] = \arg\max_{v \in V} \left[\frac{p_c \cdot \text{CLIP}(v)}{|p_c||\text{CLIP}(v)|}\right]. \tag{3}$$

**Pre-activation bias in similarity calculations.** While the method above suggests removing $b_{pre}$ from CLIP embeddings, our empirical analysis revealed this significantly masks neuron-concept relationships. Without $b_{pre}$, in Figure 9, we show that similarities cluster around 0.1 (mean) with maxima around $0.15 - 0.2$, whereas retaining $b_{pre}$ yields higher similarity scores ($>0.42$) that correspond to correct concepts. Importantly, both approaches preserve neuron rankings, with over 95% of concepts sharing identical highest matching neurons, so not removing bias doesn't destroy ranking. Manual evaluation confirmed that neurons with bias-removed similarities ($\sim 0.2$) are under-estimated compared to their bias-inclusive counterparts ($\sim 0.5$). Based on these findings, we retain $b_{pre}$ in our calculations.

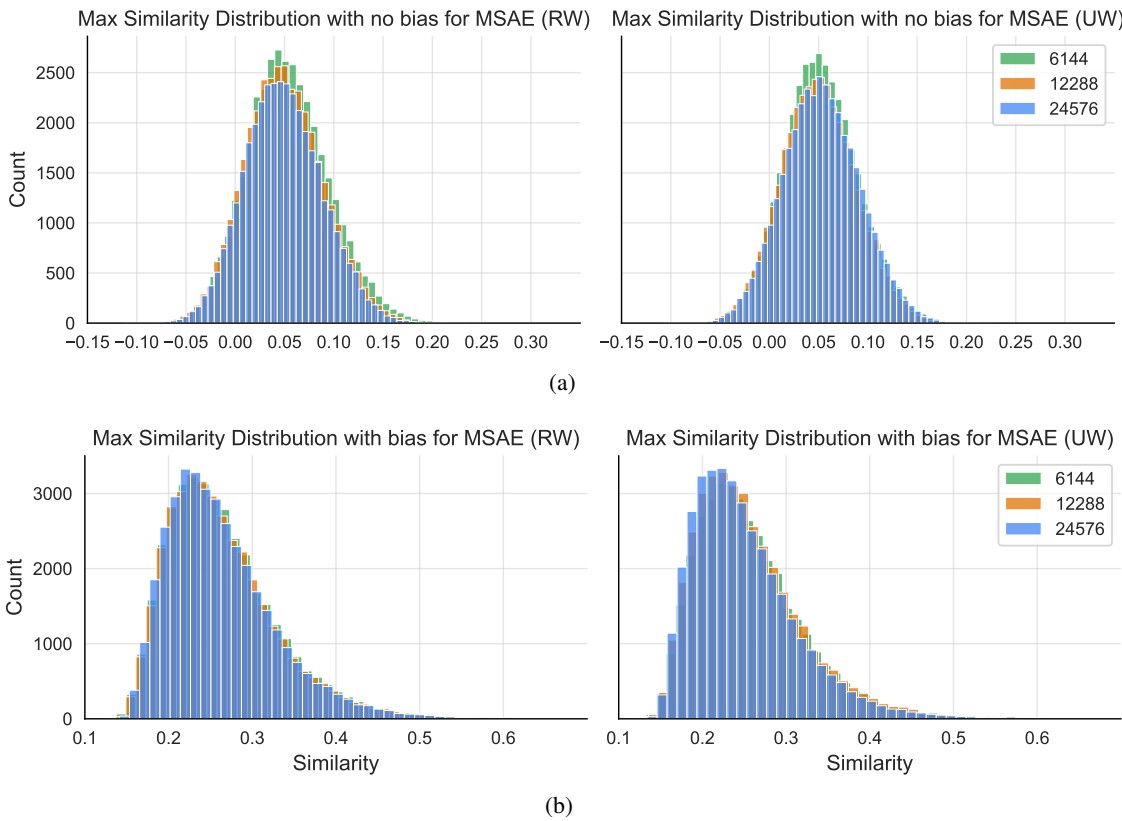

*Figure 9.* **Impact of pre-activation bias on concept similarities.** We take the highest neuron similarities per concept across expansion rates for (RW) and (UW) MSAE variants, (a) with and (b) without $b_{pre}$. Not removing $b_{pre}$ yields a better distribution with higher similarities that better reflect neuron interpretability.

**Limitations of the current approach.** We identify several limitations in the current concept mapping approach. First, the method assigns concepts to neurons based on the highest similarity regardless of the absolute matching quality, potentially leading to poor concept assignments when no good matches exist. Second, hierarchical concepts pose a challenge when matching with more specific neuronal features. For example, a high-level concept like 'mammal' may show strong similarity to both 'cat' and 'dog' neurons, resulting in imprecise assignments. This issue stems from either semantic feature vectors that aren't perfectly orthogonal or incomplete vocabulary coverage.

**Threshold-based validation.** To address the challenges identified above, we propose three validations to remove weak assignments. Before applying the validations, we switch the mapping from concepts to neurons, to a mapping of neurons to concepts to reduce spurious assignments. Based on this, we threshold results by either:

1. Cosine similarity $> 0.42$, which ensures that the neurons exhibit strong alignment with their assigned concepts, preventing weak or ambiguous concept mappings

2. Concept similarity ratio $\frac{\text{Top similarity}}{\text{Second-highest similarity}} > 2.0$, confirms concept uniqueness by requiring the best match to be at least twice as strong as the second-best concept, avoiding distributed representations

3. One concept per neuron (with the highest similarity) enforces monosemanticity by assigning only the most strongly aligned concept to each neuron, which is needed due to the vocabulary structure containing multiple variations of the same concept (e.g., 'bird' and 'birdie')

**Vocab data.** Following (Ramaswamy et al., 2023) principle that vocabulary concepts should be simple, we adopted the vocabulary from (Bhalla et al., 2024). This vocabulary comprises the most frequent unigrams from LAION-400m captions dataset (Schuhmann et al., 2021). To account for semantic relationships between concepts, we perform manual validation of top concepts for each discovered neuron.

**Semantic consistency.** Manual evaluation of top concepts per neuron verifies concept consistency and identifies hierarchical relationships, where top vocab similarities (such as dog breeds) can indicate broader categorical concepts (such as 'dog').

**Results across SAE architectures.** We evaluate concept neurons across architectures in Table 3. From 37,445 neurons at expansion rate 8, only a small fraction passed similarity validation: ~10% for TopK and 1–3% for ReLU and Matryoshka architectures. While higher expansion rates typically reduce valid neurons, both TopK variants and ReLU ($\lambda = 0.001$) exhibit increased valid mappings under best vector validation. Although these results suggest limited concept learning or concept distribution across neurons, the vocabulary structure prevents definitive conclusions, due to the dominance of non-semantic unigrams, and many semantically similar concepts appear across vocabulary (e.g., 'blue', 'blau' and 'bleu'). The validation results from the table demonstrate that sparser architectures (TopK) yield 3–8 times more interpretable concept neurons compared to more dense ones (ReLU), with Matryoshka being between the two, supporting the hypothesis that sparsity promotes concept specialization.

*Table 3.* **Comparison of valid concept neurons detected across different SAEs and validation methods.** The validation methods include a cosine similarity threshold above 0.42, selecting the best matching neuron, combining both criteria, applying the concept similarity ratio threshold between the first and second best vocab concept for the neuron, and enforcing all conditions simultaneously. Measurements were made for each model at three expansion rates ($\times 8$ | $\times 16$ | $\times 32$).

| Model | Similarity above 0.42 | Best vector | Above and best | Ratio threshold | All conditions |
|---|---|---|---|---|---|
| ReLU ($\lambda = 0.03$) | 3308 \| 3765 \| 4181 | 2740 \| 3608 \| 5129 | 874 \| 1046 \| 1304 | 380 \| 175 \| 45 | 97 \| 31 \| 16 |
| ReLU ($\lambda = 0.003$) | 896 \| 781 \| 799 | 2372 \| 3305 \| 5129 | 217 \| 196 \| 188 | 395 \| 251 \| 194 | 29 \| 19 \| 7 |
| ReLU ($\lambda = 0.001$) | 351 \| 247 \| 128 | 4116 \| 6793 \| 11417 | 77 \| 63 \| 32 | 169 \| 47 \| 3 | 8 \| 2 \| 0 |
| TopK ($k = 32$) | 4081 \| 4719 \| 5027 | 2755 \| 3415 \| 3827 | 1021 \| 1259 \| 1411 | 999 \| 857 \| 858 | 216 \| 197 \| 203 |
| TopK ($k = 64$) | 3797 \| 4504 \| 4915 | 2557 \| 3272 \| 167 | 873 \| 1080 \| 1238 | 1322 \| 1151 \| 1167 | 238 \| 232 \| 238 |
| TopK ($k = 128$) | 2141 \| 2590 \| 3059 | 2167 \| 2670 \| 3306 | 455 \| 565 \| 745 | 1508 \| 1383 \| 1379 | 211 \| 226 \| 231 |
| TopK ($k = 256$) | 943 \| 888 \| 962 | 1883 \| 2191 \| 2631 | 168 \| 167 \| 171 | 1579 \| 1523 \| 1554 | 134 \| 126 \| 127 |
| Matryoshka (RW) | 1136 \| 1109 \| 1038 | 1628 \| 2213 \| 2541 | 237 \| 257 \| 259 | 1429 \| 1135 \| 1059 | 140 \| 132 \| 121 |
| Matryoshka (UW) | 907 \| 894 \| 748 | 1517 \| 1908 \| 2396 | 195 \| 191 \| 167 | 1254 \| 1169 \| 1069 | 125 \| 128 \| 98 |

# B. Implementation Details

We conducted experiments using CLIP ViT-L/14 (and ViT-B/16, reported later in the appendix) pre-trained on the CC3M dataset image training subset. Following (Bricken et al., 2023; Gao et al., 2025) and this blog (Conerly et al., 2024), our SAE implementation uses unit-norm constraint on the decoder columns with untied encoder and decoder. We initialized $b_{\text{pre}}$ and $b_{\text{enc}}$ with zeros, decoder with uniform Kaiming initialization (scaled $L_2$ norm to 0.1), and encoder as the decoder's transpose. Gradient clipping was set to 1. For data preprocessing, we centralized embeddings per modality (Bhalla et al., 2024) and scaled by a constant to achieve $\mathbb{E}_{x \in \mathcal{X}}[\|x\|_2] = \sqrt{n}$. All models were trained for 30 epochs on a single NVIDIA A100 GPU with batch size 4096, except for the model with an expansion rate of 32, which was trained for 20 epochs. While MSAE and TopK showed dead neurons, we omitted revival strategies as only in TopK the number of dead neurons exceeded 1%.

## B.1. Hyperparameters

We first conducted experiments on CLIP RN50 using hyperparameters from (Rao et al., 2024), later validating them on ViT-L/14. For ViT-L/14, we explored parameters near RN50-optimal values to ensure cross-architecture consistency. With expansion factor 8 ($768 \rightarrow 6144$), we explore:

- Learning rates per method: $1 \cdot 10^{-5}, 5 \cdot 10^{-5}, 1 \cdot 10^{-4}, 5 \cdot 10^{-4}, 1 \cdot 10^{-3}$

- ReLU $L_1$ coefficients ($\lambda$): $1 \cdot 10^{-4}, 3 \cdot 10^{-3}, 1 \cdot 10^{-3}, 3 \cdot 10^{-2}$

- TopK values: k $\in \{32, 64, 128, 256\}$, up to 256 as (Gao et al., 2025) suggests higher values do not learn interpretable features

- Matryoshka K-lists: $\{32 \ldots 6144\}$ and $\{64 \ldots 6144\}$, for higher expansion rates we adjust the upper limit

- $\alpha$ coefficients: uniform weighting (UW) $\{1,1,1,1,1,1,1\}$ and reverse weighting (RW) $\{7,6,5,4,3,2,1\}$

The optimal parameters from these experiments were applied to larger expansion factors of 16 ($768 \rightarrow 12288$), 32 ($768 \rightarrow 24576$), and all expansion rates of VIT-B/16.

Following reviewer suggestions, we extended our evaluations to include TopK ($k = 512$) and BatchTopK for $\in \{16, 32, 64, 128, 256\}$. The BatchTopK models were trained using the same hyperparameters as TopK models. These additional results are presented in the Appendix. We also attempted to integrate JumpReLU (Rajamanoharan et al., 2024b) into our evaluations, but did not achieve meaningful results, which may be attributed to an implementation error in our code.

## B.2. Optimal Parameters

Based on RN50 experiments and subsequent adjustment to VIT-L/14 with expansion factor 8, we selected the following optimal configurations: ReLU with learning rate $5 \cdot 10^{-5}$ and $\lambda$ values of $1 \cdot 10^{-3}, 3 \cdot 10^{-3}, 3 \cdot 10^{-2}$; TopK with learning rate $5 \cdot 10^{-4}$ and k values of 32, 64, 128, and 256; MSAE with learning rate $1 \cdot 10^{-4}$, K-list $\{64 \ldots 6144\}$, for both uniform (UW) and reverse weighting (RW) $\alpha$ strategies.

## C. Highest Neuron Magnitudes

Based on results from Figure 4, we analyze images from ImageNet-1k validation set that produced the highest neuron magnitudes for TopK and MSAE architectures. In Table 4, we show that more constrained SAEs (TopK ($k \leq 128$)) produce a higher number of samples with neurons above 15; however, the percentage of valid neurons is lower than in MSAE and TopK ($k = 256$) which have significantly less high magnitude samples. This indicates that high-magnitude neurons in highly constrained TopK may presumably learn complex features. Figure 10 presents the top 6 valid highest neuron magnitude images per model, demonstrating that very high magnitudes often correspond to images with almost singular concepts.

*Table 4.* **Analysis of high-magnitude neurons across architectures.** We analyze samples with magnitude $> 15$ in the ImageNet-1k validation set, showing the number of total occurrences, the proportion of valid concepts among high-magnitude concepts, and the rate of high-magnitude valid concepts relative to all valid concepts in the model from Table 3.

| Model | High-Magnitude Samples | Valid Concept Rate | High-Magnitude Concept Rate |
|---|---|---|---|
| TopK ($k = 32$) | 113 | 6 (5%) | 216 (3%) |
| TopK ($k = 64$) | 18 | 0 (0%) | 238 (0%) |
| TopK ($k = 128$) | 3 | 0 (0%) | 211 (0%) |
| TopK ($k = 256$) | 12 | 8 (67%) | 134 (6%) |
| MSAE (RW) | 21 | 8 (38%) | 140 (6%) |
| MSAE (UW) | 22 | 7 (32%) | 125 (6%) |

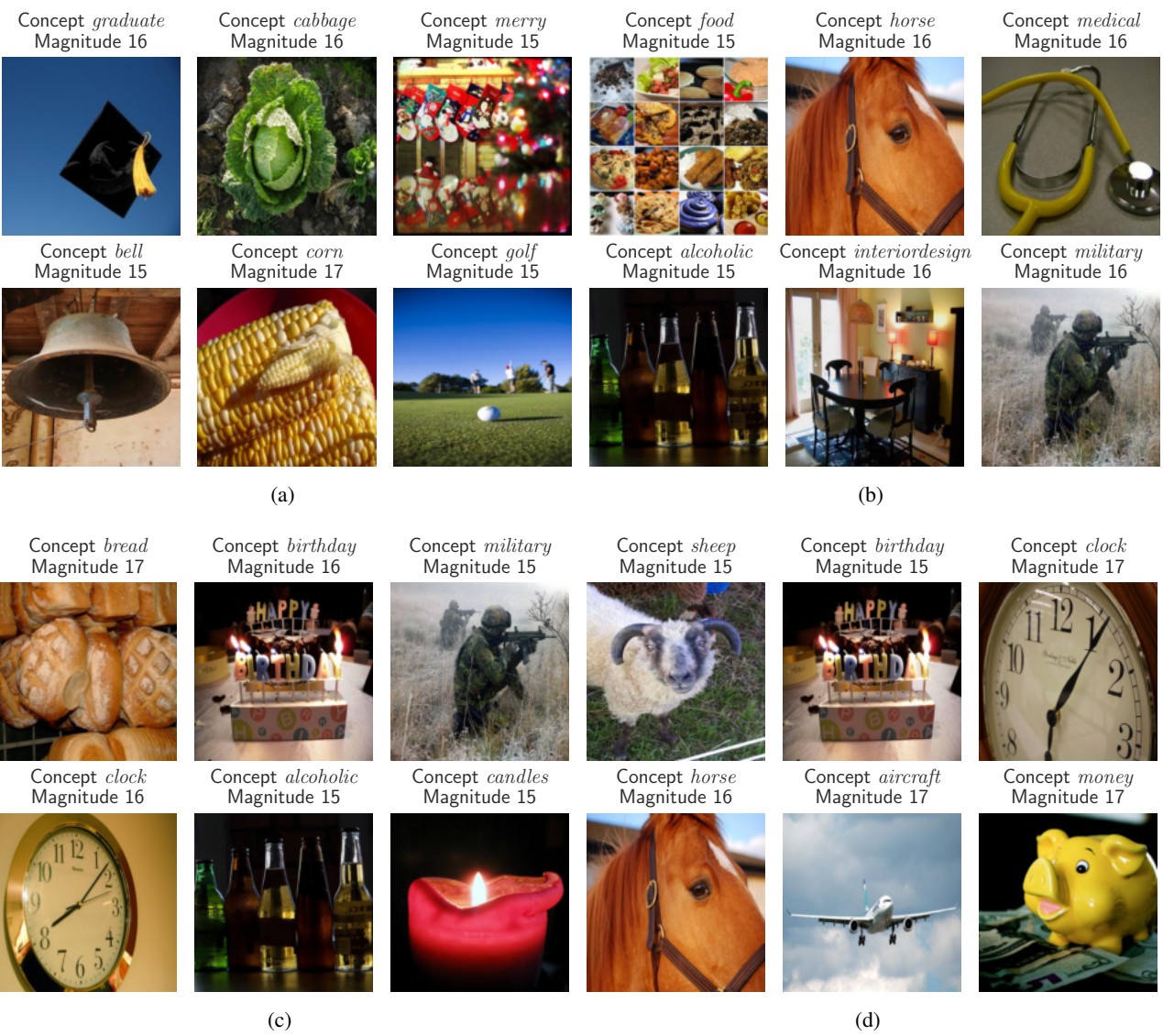

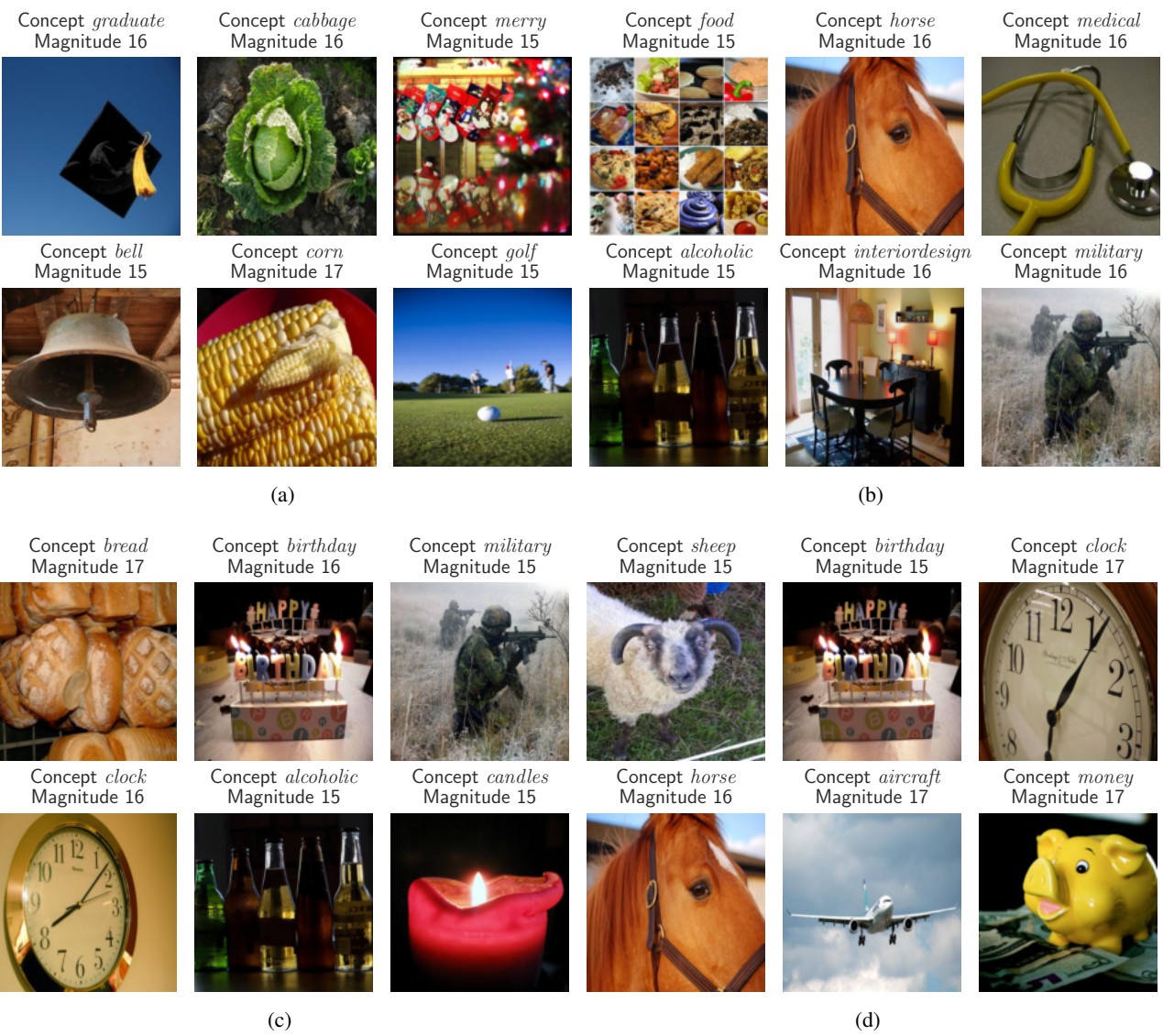

*Figure 10.* **Images with the highest valid concept neuron magnitudes.** We took 6 images from ImageNet-1k validation set per model based on results from Table 4 with (a) TopK $k = 32$, (b) TopK $k = 256$, (c) Matryoshka RW, and (d) Matryoshka UW.

# D. Activation Soft-capping

Analysis of MSAE in Figure 4 reveals that despite effective handling of multi-granular sparsity, the model learns to encode concepts using extremely large activation values ($>15$). This can lead to more composite rather than atomic features, as it was in the case of TopK ($k \leq 128$) revealed in Appendix C.

## D.1. Definition of Soft-capping.

To address this, we introduce activation soft-capping (SC), adapting the logit soft-capping concept from language models (Gemma & DeepMind, 2024). This technique prevents too high activation magnitude and circumvention of sparsity constraints via activation magnitude manipulation:

$$\hat{z} = \text{softcap} \cdot \tanh(z/\text{softcap}), \quad \hat{x} = W_{\text{dec}}\hat{z} + b_{\text{pre}}, \tag{4}$$

where $\text{softcap}$ hyperparameter controls maximum activation magnitude. Combined with ReLU, this bounds SAE activations to $(0, \text{softcap})$.

## D.2. Results.

In Table 5, we show soft-capping's impact on MSAE performance across key metrics using the ImageNet-1k training set. Our analysis reveals two key benefits of applying soft-capping on MSAE. First, it consistently improves $L_0$ sparsity, with MSAE RW (SC) achieving values of 0.830 and 0.889 for 6144 and 12288 sizes, respectively. Second, while base MSAE UW maintains better FVU and CS scores, the soft-capped MSAE RW significantly reduces the number of dead neurons. With a latent size of 12288, MSAE RW exhibits only 66 dead neurons, compared to 491 in the model with a latent size of 6144. These findings show that soft-capping is particularly beneficial for large-scale SAEs with wider sparse layers, where neuron utilization becomes more challenging. The technique provides a practical approach to reducing dead neurons while maintaining high $L_0$ sparsity, with only minimal impact on reconstruction fidelity.

*Table 5.* **Impact of soft-capping (SC) on MSAE performance.** We evaluate soft-capping across different expansion rates (8 and 16) on the ImageNet-1k validation set, comparing UW and RW variants. While base MSAE maintains better FVU and CS scores, soft-capped variants show improved $L_0$ sparsity and reduced number of dead neurons, particularly at larger sizes. **Bold** values indicate the best performance per metric and size, with NDN in parentheses showing dead neuron counts from the final checkpoint.

| Size | Model | $L_0 \uparrow$ | FVU $\downarrow$ | CS $\uparrow$ | CKNNA $\uparrow$ | NDN $\downarrow$ |
|---|---|---|---|---|---|---|
| 6144 | Matryoshka (RW) | $0.829_{\pm.008}$ | $0.007_{\pm.003}$ | $0.997_{\pm.002}$ | $0.809_{\pm.002}$ | 2(4) |
| | Matryoshka (UW) | $0.748_{\pm.006}$ | $\mathbf{0.001}_{\pm.002}$ | $\mathbf{0.999}_{\pm.000}$ | $0.848_{\pm.003}$ | 0(22) |
| | Matryoshka (RW, SC) | $\mathbf{0.830}_{\pm.007}$ | $0.010_{\pm.003}$ | $0.995_{\pm.002}$ | $0.839_{\pm.004}$ | **1(2)** |
| | Matryoshka (UW, SC) | $0.774_{\pm.006}$ | $0.004_{\pm.001}$ | $0.998_{\pm.001}$ | $\mathbf{0.856}_{\pm.003}$ | 1(3) |
| 12288 | Matryoshka (RW) | $0.884_{\pm.006}$ | $0.005_{\pm.003}$ | $0.998_{\pm.001}$ | $0.801_{\pm.003}$ | 32(124) |
| | Matryoshka (UW) | $0.830_{\pm.003}$ | $\mathbf{0.000}_{\pm.000}$ | $\mathbf{1.000}_{\pm.000}$ | $\mathbf{0.853}_{\pm.002}$ | 22(491) |
| | Matryoshka (RW, SC) | $\mathbf{0.889}_{\pm.005}$ | $0.007_{\pm.003}$ | $0.997_{\pm.001}$ | $0.833_{\pm.002}$ | **11(66)** |
| | Matryoshka (UW, SC) | $0.842_{\pm.005}$ | $0.001_{\pm.001}$ | $0.999_{\pm.000}$ | $0.849_{\pm.002}$ | 87(172) |

# E. CKNNA Alignment Metric

Introduced in Section 4.4, CKNNA (Centered Kernel Nearest-Neighbor Alignment) measures representation similarity between networks while focusing on local neighborhood structures. Unlike its predecessor CKA (Kornblith et al., 2019), CKNNA refines the alignment computation by considering only $k$-nearest neighbors, making it more sensitive to local geometric relationships. The alignment score between two networks' representations in our case CLIP embeddings and SAE activation is computed as:

$$\text{CKNNA}(\mathbf{K}, \mathbf{L}) = \frac{\text{Align}(\mathbf{K}, \mathbf{L})}{\sqrt{\text{HSIC}(\mathbf{K}, \mathbf{K})\text{HSIC}(\mathbf{L}, \mathbf{L})}},$$

$$\text{HSIC}(\mathbf{K}, \mathbf{L}) = \frac{1}{(n-1)^2} \left( \sum_i \sum_j (\langle \phi_i, \phi_j \rangle - \mathbb{E}_l[\langle \phi_i, \phi_l \rangle])(\langle \psi_i, \psi_j \rangle - \mathbb{E}_l[\langle \psi_i, \psi_l \rangle]) \right),$$

$$\text{Align}(\mathbf{K}, \mathbf{L}) = \frac{1}{(n-1)^2} \left( \sum_i \sum_j \alpha(i,j)(\langle \phi_i, \phi_j \rangle - \mathbb{E}_l[\langle \phi_i, \phi_l \rangle])(\langle \psi_i, \psi_j \rangle - \mathbb{E}_l[\langle \psi_i, \psi_l \rangle]) \right),$$

$$\alpha(i,j;k) = \mathbf{1}[i \neq j \text{ and } \phi_j \in \text{knn}(\phi_i; k) \text{ and } \psi_j \in \text{knn}(\psi_i; k)],$$

(5)

where HSIC measures the global similarity between kernel matrices, and Align introduces the neighborhood constraint through $\alpha(i,j;k)$. The indicator function $\alpha(i,j;k)$ ensures that only pairs of points that are $k$-nearest neighbors in both representation spaces contribute to the alignment score. Here, $\phi_i, \phi_j$ represent CLIP embeddings and $\psi_i, \psi_j$ represent SAE activations for corresponding input data points $i$ and $j$. Following (Yu et al., 2025), we set $k = 10$ as it provides better alignment sensitivity and calculate CKNNA over randomly sampled (batch size) 10,000 representations when evaluating. Higher CKNNA scores indicate stronger similarity between the CLIP and SAE learned representations.

# F. Evaluating MSAE: Additional Results

We extend the results from Section 4 by analyzing multiple expansion rates, SAE variants, input modalities, and CLIP architectures. Unless otherwise specified, experiments use CLIP ViT-L/14 with an expansion rate of 8 on image modality. For text modality evaluations, we use the CC3M validation subset, while image modality evaluations are performed on the ImageNet-1k training subset.

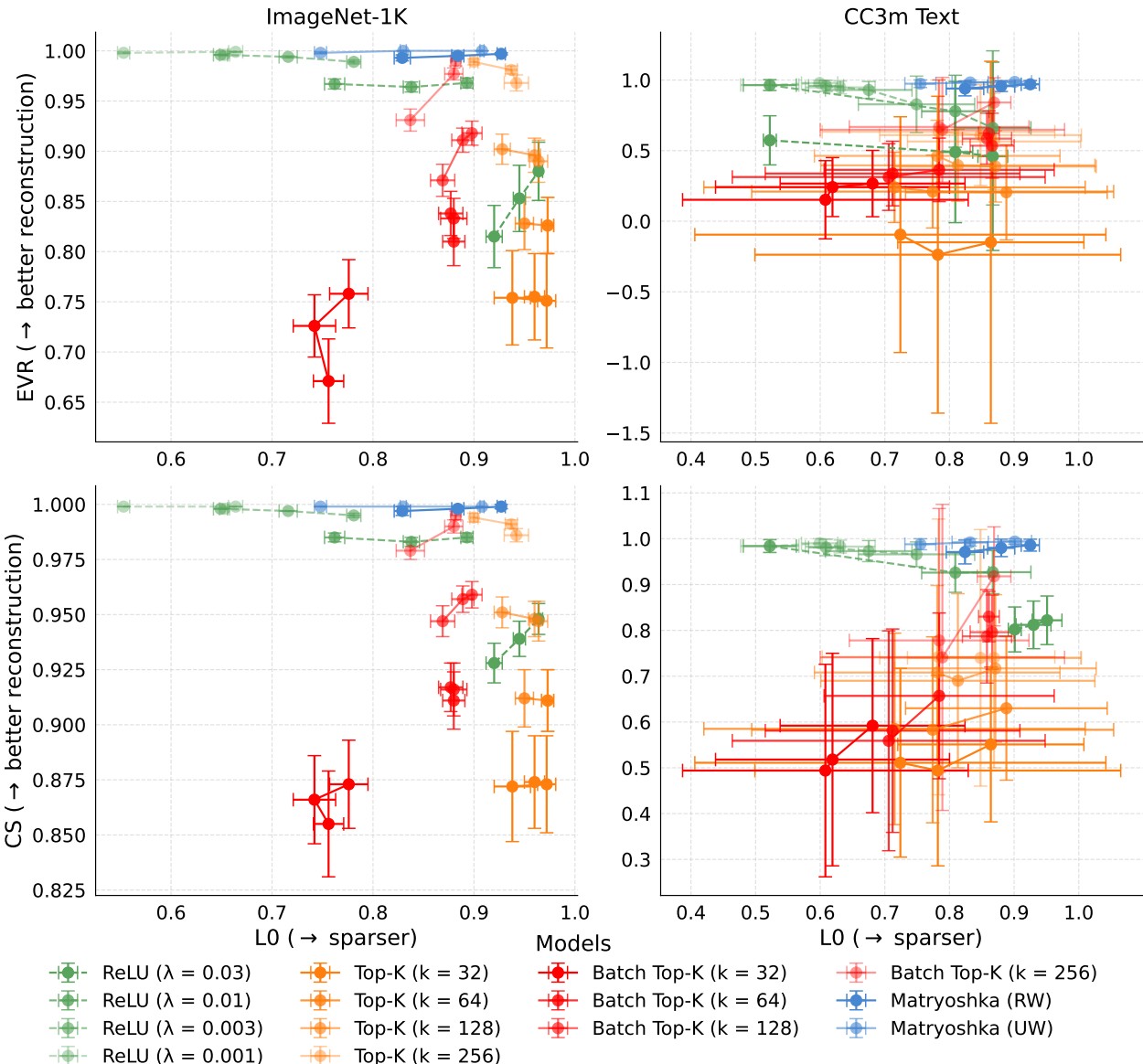

*Figure 11.* **Extended sparsity-fidelity trade-off analysis across modalities.** Expanding on Figure 2, we compare ReLU SAE ($\lambda =$ 0.03, 0.01, 0.003, 0.001), TopK SAE ($k = 32, 64, 128, 256$), BatchTopK SAE ($k = 32, 64, 128, 256$), and MSAE (RW, UW) using two reconstruction metrics: mean EVR fidelity (top) and mean cosine similarity (bottom). Results are shown for both image (left) and text (right) modalities, with standard deviation also reported for each metric, demonstrating MSAE's consistent performance across modalities and metrics.

## F.1. Sparsity–Fidelity Trade-off

Figure 11 presents an extended analysis of sparsity-fidelity trade-offs, including standard deviations and an alternative reconstruction metric tailored for CLIP embeddings (cosine similarity). The results demonstrate MSAE's superior stability

across both modalities, particularly in text representations where only MSAE models shows stable and elevated results. Furthermore, we observe that models learned with lower $k$ or higher sparsity regularization show high variance on the trained (image) modality, and this instability becomes even more pronounced on the text modality. Due to MSAE's inherently low variance in the image modality, this similar low variance is preserved in the other modality, leading to consistently high and stable performance across both modalities. Figure 12 further strengthens our findings by showing MSAE's superiority on a different CLIP architecture (ViT-B/16).

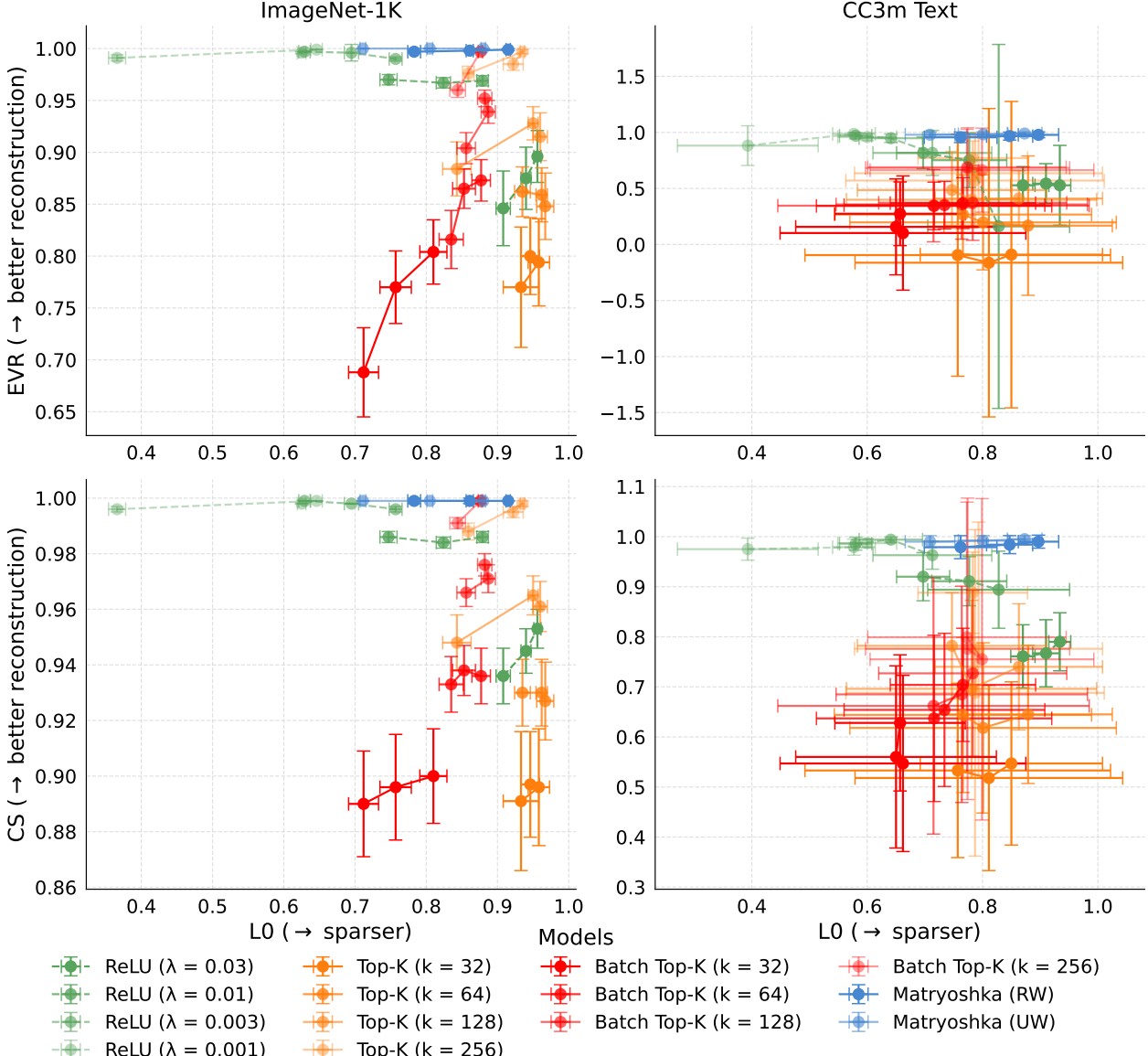

*Figure 12.* **ViT-B/16 sparsity-fidelity trade-off analysis across modalities.** Parallel analysis to ViT-L/14 (Figure 11), demonstrating that MSAE's superior performance and stability generalizes across CLIP architectures.

## F.2. Ablation: Matryoshka at Lower Granularity Levels

Figure 13 extends the analysis from Figure 3 by evaluating four key metrics: reconstruction fidelity (EVR), reconstruction error (CS), alignment (CKNNA), and neuron utilization (NDN). Our expanded comparison reinforces MSAE's competitive performance against TopK SAE, demonstrating that MSAE (RW) achieves similar or better results across most metrics except for NDN, where (UW) version of MSAE performs better.

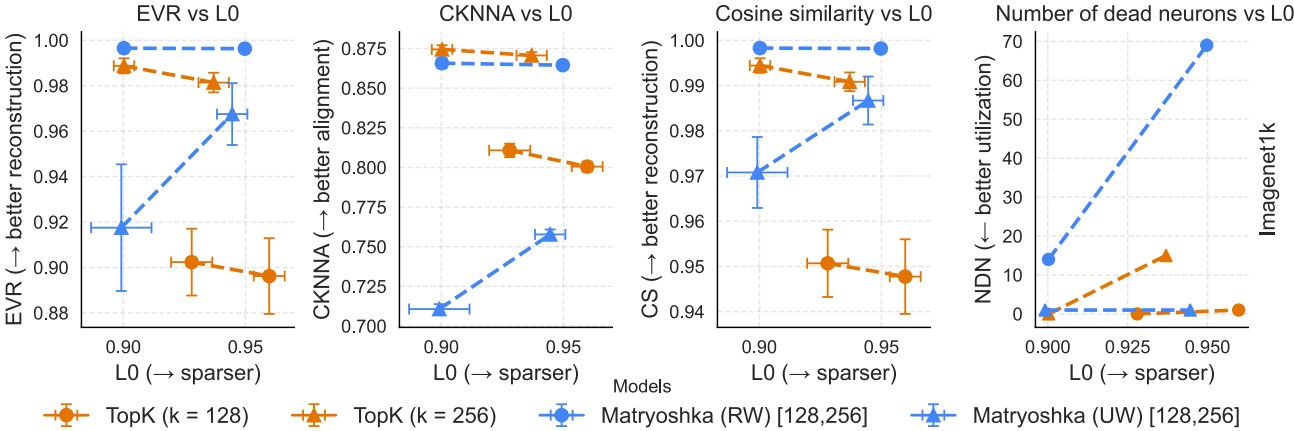

*Figure 13.* **Comprehensive comparison of Matryoshka and TopK SAE on ImageNet-1k.** Extension of Figure 3 comparing model performance on reconstruction fidelity (EVR), reconstruction error (CS), concept alignment (CKNNA), and neuron utilization (NDN) against sparsity (L0). MSAE (RW) demonstrates competitive performance across most metrics, while MSAE (UW) achieves better results in neuron utilization.

## F.3. Activation Magnitudes Analysis

We extend the activation magnitude analysis from Figure 4 by including varying versions of model sparsity from each evaluated SAE architecture. Figure 14 shows non-zero and maximum SAE activations for expansion rate 8, revealing that less constrained TopK models exhibit double-curvature distributions similar to MSAE and ReLU. Maximum activation analysis highlights ReLU's shrinkage effect, while TopK and MSAE maintain distributions closer to normal. These patterns persist at higher expansion rates (16 and 32) as shown in Figure 15.

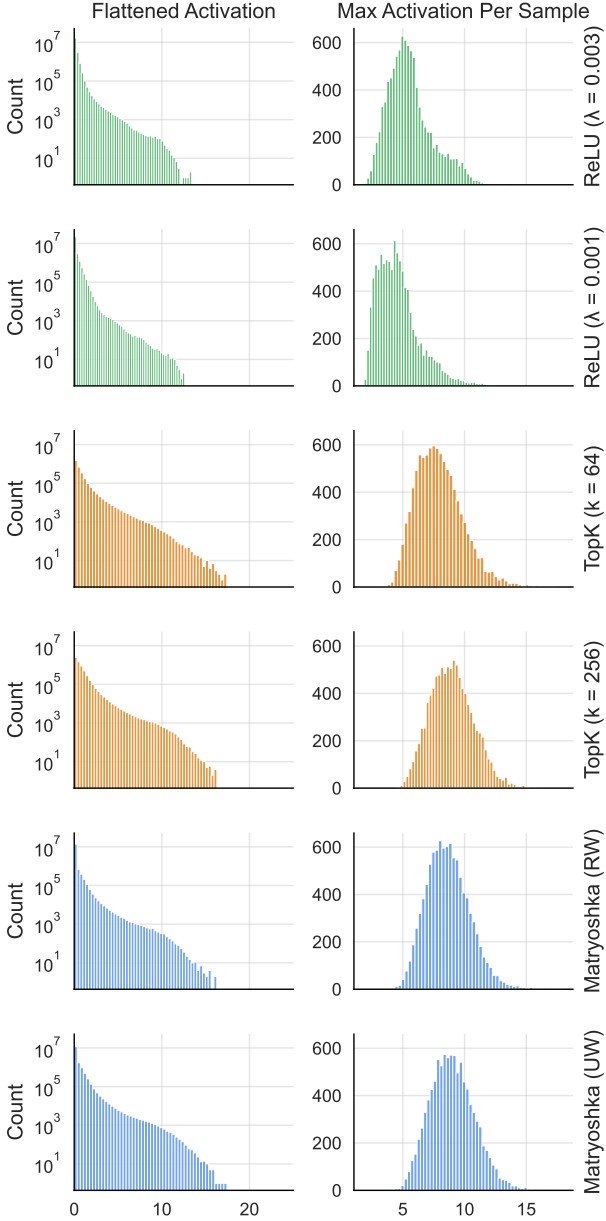

Figure 14. **Activation distributions at expansion rate 8.** Extended analysis of Figure 4 showing: (left) non-zero activation distributions, revealing TopK's convergence to double-curvature patterns at lower constraints (higher $k$), (right) maximum activation distributions, demonstrating ReLU shrinkage problem compared to TopK and MSAE behavior which resembles a normal distribution.

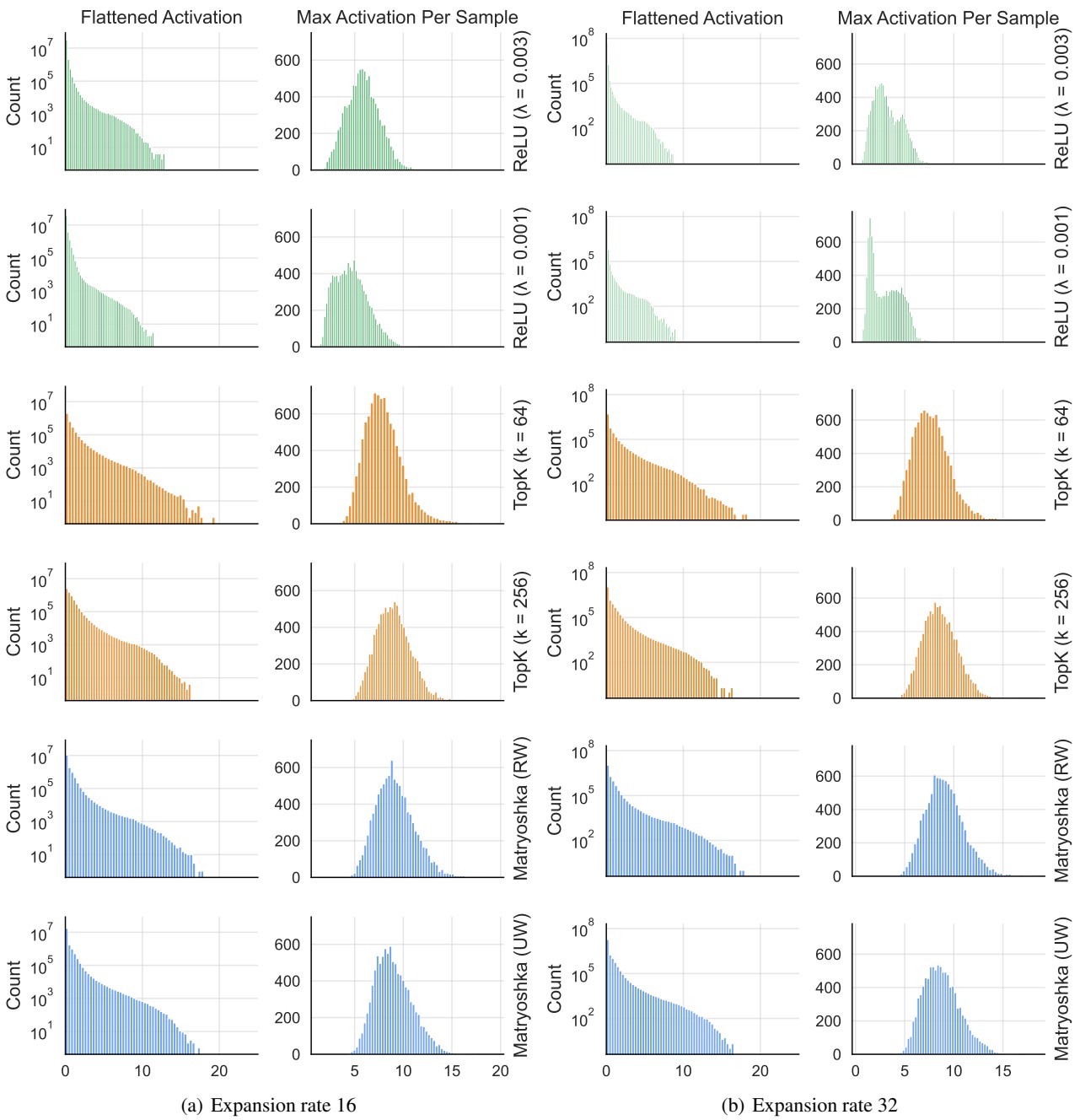

*Figure 15.* **Activation distributions at higher expansion rates.** Extended analysis of Figure 4 for expansion rates 16 (a) and 32 (b), showing consistency of distribution patterns across scales.

## F.4. Progressive Recovery

We extend the analysis from Figure 2 by examining progressive reconstruction performance across additional metrics and modalities. Figure 16 demonstrates performance at expansion rate 8 for reconstruction quality (EVR, CS) and neuron utilization (NDN) across both modalities, while Figure 17 extends this analysis to expansion rates 16 and 32.

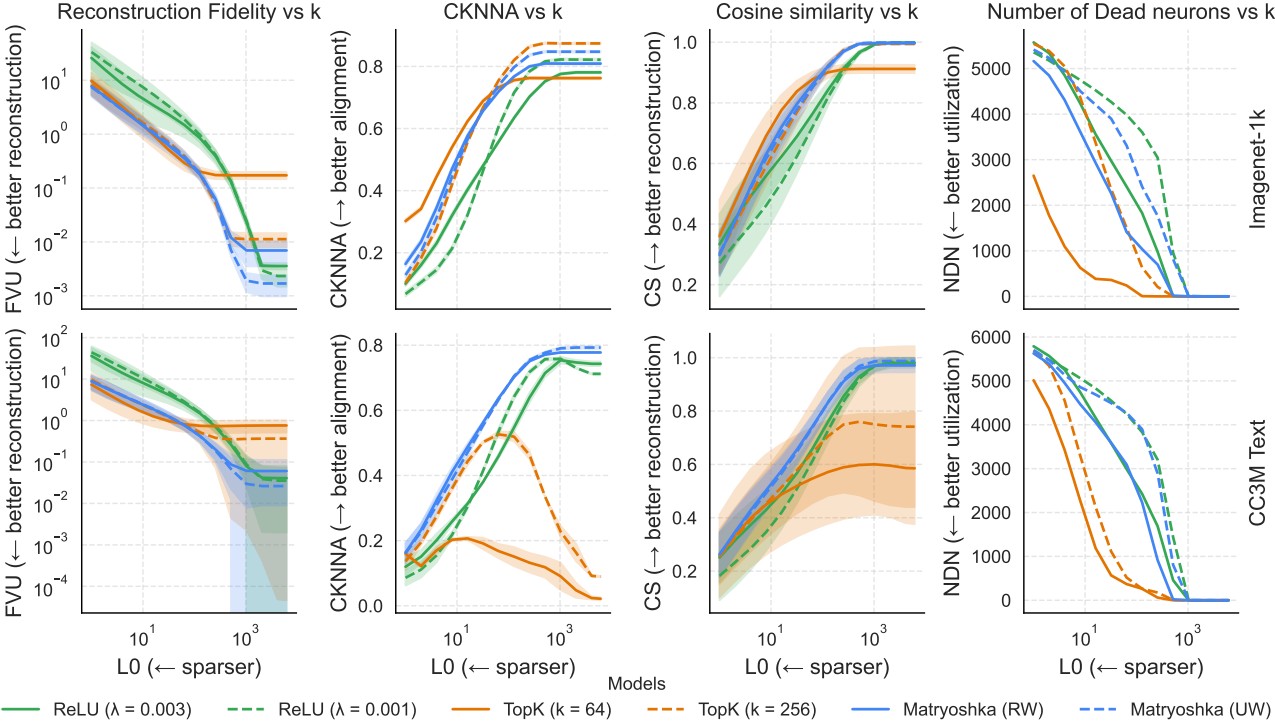

*Figure 16.* **Progressive recovery analysis at expansion rate 8.** Extension of Figure 2 showing reconstruction (EVR, CS), alignment (CKNNA), and neuron utilization (NDN) metrics against an increasing number of utilized top magnitude SAE neurons for image and text modalities. MSAE demonstrates comparable performance to TopK ($k = 256$) on image modality and superior performance on text.

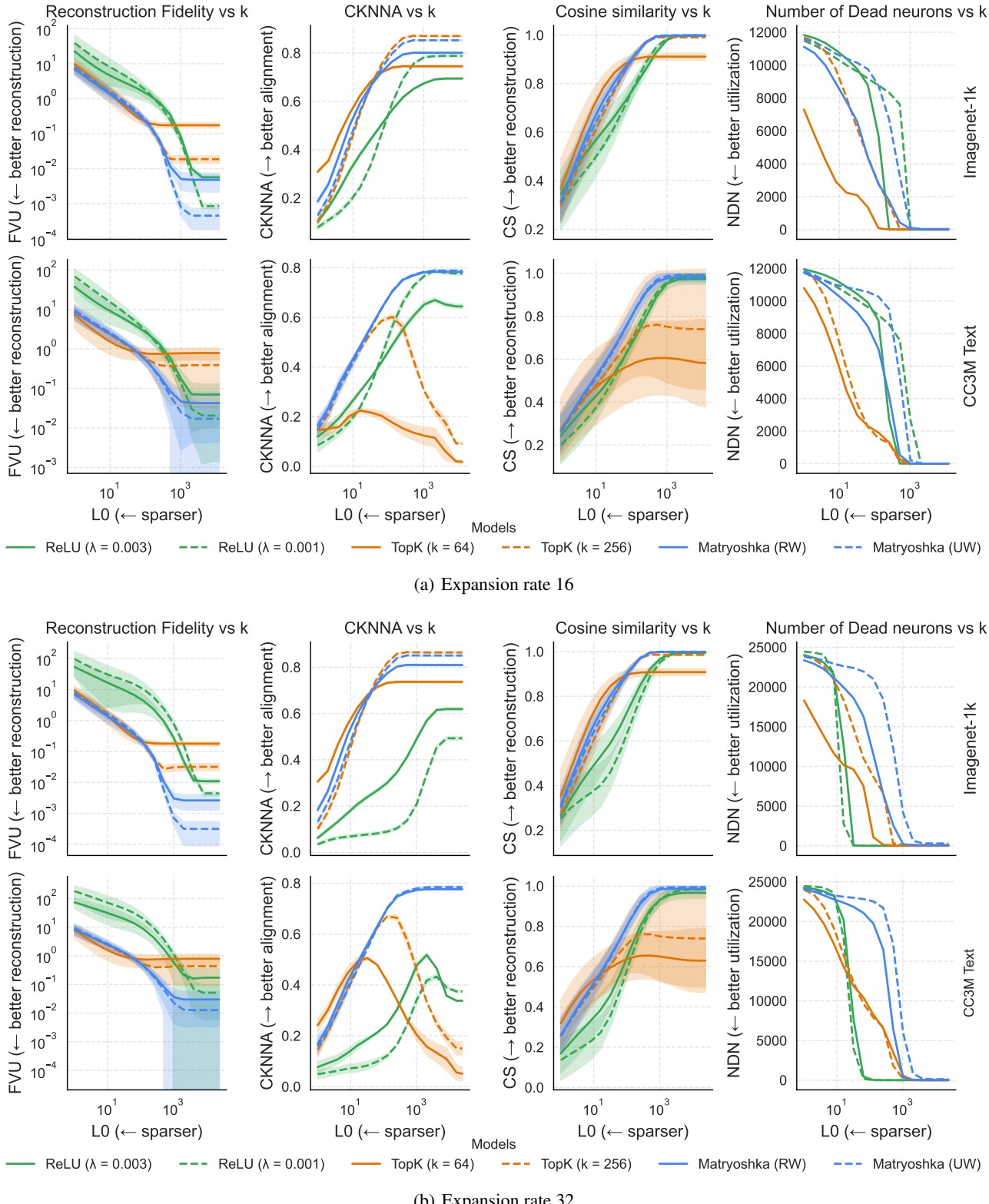

(a) Expansion rate 16

(b) Expansion rate 32

*Figure 17.* **Progressive recovery analysis at higher expansion rates.** Analysis parallel to Figure 16 for expansion rates 16 (a) and 32 (b), demonstrating the stability of the results over higher expansion rates.

### F.5. Comprehensive Evaluation with CLIP ViT-L/14 and ViT-B/16 Architectures

We present an extensive quantitative comparison of SAE variants across CLIP ViT-L/14 and ViT-B/16 architectures. Our evaluation encompasses ReLU ($\lambda = 0.03, 0.01, 0.003, 0.001$), TopK ($k = 32, 64, 128, 256, 512$), BatchTopK ($k = 16, 32, 64, 128, 256$), and MSAE (RW, UW) models, tested across three expansion rates (8, 16, 32) for both image and text modalities. For text modality and ViT-B/16 architecture, we omit LP (Acc) and LP (KL) metrics based on our findings in Section 4.4 that CS and FVU correlate strongly with linear probing metrics.

#### F.5.1. RESULTS FOR IMAGE MODALITY

For image modality, Tables 6–8 present detailed results for ViT-L/14 across expansion rates 8, 16, and 32, while Tables 9–11 show parallel performance metrics for ViT-B/16. These tables extend the analysis from Table 1, providing comprehensive measurements across different metrics and model configurations.

*Table 6.* **CLIP ViT-L/14 SAE comparison at expansion rate 8.** Extended evaluation from Table 1 with additional TopK, ReLU and BatchTopK variants on ImageNet-1k. Arrows indicate preferred metric direction, NDN values show training set dead neurons in parentheses.

| Model | $L_0 \uparrow$ | FVU $\downarrow$ | CS $\uparrow$ | LP (KL) $\downarrow$ | LP (Acc) $\uparrow$ | CKNNA $\uparrow$ | DO $\downarrow$ | NDN $\downarrow$ |
|---|---|---|---|---|---|---|---|---|
| ReLU ($\lambda = 0.03$) | $.920_{\pm.008}$ | $.185_{\pm.031}$ | $.928_{\pm.009}$ | $50.5_{\pm77.1}$ | $.936_{\pm.244}$ | $.727_{\pm.004}$ | .003 | 0(0) |
| ReLU ($\lambda = 0.01$) | $.762_{\pm.010}$ | $.033_{\pm.005}$ | $.985_{\pm.002}$ | $7.16_{\pm11.3}$ | $.977_{\pm.149}$ | $.742_{\pm.005}$ | .002 | 0(0) |
| ReLU ($\lambda = 0.003$) | $.649_{\pm.007}$ | $.004_{\pm.000}$ | $.998_{\pm.000}$ | $0.66_{\pm1.03}$ | $.994_{\pm.083}$ | $.781_{\pm.004}$ | .003 | 0(0) |
| ReLU ($\lambda = 0.001$) | $.553_{\pm.006}$ | $.002_{\pm.001}$ | $.999_{\pm.000}$ | $0.36_{\pm0.65}$ | $.995_{\pm.073}$ | $.822_{\pm.004}$ | .002 | 0(0) |
| TopK ($k = 32$) | $.960_{\pm.010}$ | $.245_{\pm.043}$ | $.874_{\pm.021}$ | $109_{\pm161}$ | $.903_{\pm.300}$ | $.711_{\pm.005}$ | .002 | 0(1235) |
| TopK ($k = 64$) | $.950_{\pm.009}$ | $.172_{\pm.026}$ | $.912_{\pm.013}$ | $60.1_{\pm90.8}$ | $.930_{\pm.255}$ | $.762_{\pm.004}$ | .002 | 0(335) |
| TopK ($k = 128$) | $.928_{\pm.008}$ | $.098_{\pm.015}$ | $.951_{\pm.007}$ | $2.71_{\pm5.40}$ | $.987_{\pm.114}$ | $.811_{\pm.004}$ | .003 | 0(117) |
| TopK ($k = 256$) | $.900_{\pm.004}$ | $.011_{\pm.003}$ | $.994_{\pm.002}$ | $2.71_{\pm5.40}$ | $.987_{\pm.114}$ | $.874_{\pm.003}$ | .003 | 0(296) |
| TopK ($k = 512$) | $.922_{\pm.015}$ | $.346_{\pm.442}$ | $.923_{\pm.058}$ | $56.1_{\pm13.9}$ | $.950_{\pm.218}$ | $.006_{\pm.003}$ | .002 | 0(1) |
| BatchTopK ($k = 16$) | $.698_{\pm.021}$ | $.371_{\pm.060}$ | $.798_{\pm.037}$ | $281_{\pm326}$ | $.836_{\pm.372}$ | $.698_{\pm.037}$ | .002 | 0(4278) |
| BatchTopK ($k = 32$) | $.776_{\pm.019}$ | $.242_{\pm.034}$ | $.873_{\pm.020}$ | $113_{\pm157}$ | $.901_{\pm.299}$ | $.735_{\pm.004}$ | .002 | 0(3080) |
| BatchTopK ($k = 64$) | $.877_{\pm.012}$ | $.162_{\pm.022}$ | $.917_{\pm.011}$ | $56.9_{\pm85.8}$ | $.931_{\pm.253}$ | $.769_{\pm.004}$ | .002 | 0(1477) |
| BatchTopK ($k = 128$) | $.898_{\pm.010}$ | $.082_{\pm.012}$ | $.959_{\pm.006}$ | $23.3_{\pm36.5}$ | $.959_{\pm.197}$ | $.805_{\pm.004}$ | .003 | 0(539) |
| BatchTopK ($k = 256$) | $.882_{\pm.005}$ | $.010_{\pm.005}$ | $.995_{\pm.002}$ | $2.42_{\pm5.12}$ | $.988_{\pm.108}$ | $.860_{\pm.003}$ | .002 | 3(919) |
| Matryoshka (RW) | $.829_{\pm.008}$ | $.007_{\pm.003}$ | $.997_{\pm.002}$ | $3.13_{\pm7.08}$ | $.987_{\pm.115}$ | $.809_{\pm.002}$ | .002 | 2(4) |
| Matryoshka (UW) | $.748_{\pm.006}$ | $.002_{\pm.001}$ | $.999_{\pm.000}$ | $0.35_{\pm0.82}$ | $.995_{\pm.070}$ | $.848_{\pm.003}$ | .001 | 0(22) |

*Table 7.* **CLIP ViT-L/14 SAE comparison at expansion rate 16.** Results parallel to Table 6 showing performance scaling at higher expansion rate on ImageNet-1k.

| Model | $L_0 \uparrow$ | FVU $\downarrow$ | CS $\uparrow$ | LP (KL) $\downarrow$ | LP (Acc) $\uparrow$ | CKNNA $\uparrow$ | DO $\downarrow$ | NDN $\downarrow$ |
|---|---|---|---|---|---|---|---|---|
| ReLU ($\lambda = 0.03$) | $.945_{\pm.006}$ | $.147_{\pm.033}$ | $.939_{\pm.008}$ | $41.1_{\pm64.8}$ | $.945_{\pm.229}$ | $.714_{\pm.004}$ | .003 | 0(0) |
| ReLU ($\lambda = 0.01$) | $.838_{\pm.008}$ | $.036_{\pm.005}$ | $.983_{\pm.002}$ | $8.41_{\pm13.7}$ | $.975_{\pm.157}$ | $.738_{\pm.005}$ | .002 | 0(0) |
| ReLU ($\lambda = 0.003$) | $.716_{\pm.009}$ | $.006_{\pm.001}$ | $.997_{\pm.000}$ | $1.08_{\pm1.74}$ | $.991_{\pm.093}$ | $.695_{\pm.003}$ | .002 | 0(0) |
| ReLU ($\lambda = 0.001$) | $.664_{\pm.007}$ | $.001_{\pm.000}$ | $.999_{\pm.000}$ | $0.14_{\pm0.22}$ | $.997_{\pm.056}$ | $.789_{\pm.004}$ | .002 | 0(0) |
| TopK ($k = 32$) | $.972_{\pm.009}$ | $.249_{\pm.047}$ | $.873_{\pm.022}$ | $112_{\pm168}$ | $.899_{\pm.301}$ | $.692_{\pm.005}$ | .002 | 0(4727) |
| TopK ($k = 64$) | $.973_{\pm.006}$ | $.174_{\pm.028}$ | $.911_{\pm.014}$ | $61.7_{\pm96.8}$ | $.927_{\pm.260}$ | $.745_{\pm.003}$ | .002 | 0(2079) |
| TopK ($k = 128$) | $.960_{\pm.006}$ | $.104_{\pm.017}$ | $.948_{\pm.008}$ | $30.7_{\pm49.0}$ | $.951_{\pm.215}$ | $.801_{\pm.003}$ | .002 | 1(897) |
| TopK ($k = 256$) | $.937_{\pm.006}$ | $.019_{\pm.004}$ | $.991_{\pm.002}$ | $3.76_{\pm6.85}$ | $.984_{\pm.127}$ | $.871_{\pm.002}$ | .004 | 15(1383) |
| TopK ($k = 512$) | $.964_{\pm.008}$ | $.336_{\pm.413}$ | $.926_{\pm.064}$ | $72.9_{\pm17.8}$ | $.944_{\pm.230}$ | $.007_{\pm.003}$ | .002 | 0(29) |
| BatchTopK ($k = 16$) | $.669_{\pm.021}$ | $.404_{\pm.055}$ | $.786_{\pm.038}$ | $310_{\pm353}$ | $.829_{\pm.377}$ | $.705_{\pm.005}$ | .001 | 0(9859) |
| BatchTopK ($k = 32$) | $.742_{\pm.021}$ | $.274_{\pm.031}$ | $.866_{\pm.020}$ | $122_{\pm166}$ | $.897_{\pm.304}$ | $.736_{\pm.005}$ | .002 | 0(8016) |
| BatchTopK ($k = 64$) | $.880_{\pm.013}$ | $.167_{\pm.020}$ | $.916_{\pm.012}$ | $55.9_{\pm85.0}$ | $.932_{\pm.252}$ | $.759_{\pm.004}$ | .002 | 0(5113) |
| BatchTopK ($k = 128$) | $.889_{\pm.011}$ | $.089_{\pm.012}$ | $.957_{\pm.006}$ | $24.6_{\pm38.5}$ | $.956_{\pm.204}$ | $.806_{\pm.003}$ | .002 | 1(2967) |
| BatchTopK ($k = 256$) | $.880_{\pm.009}$ | $.023_{\pm.006}$ | $.990_{\pm.003}$ | $4.12_{\pm7.11}$ | $.983_{\pm.131}$ | $.854_{\pm.003}$ | .002 | 12(3558) |
| Matryoshka (RW) | $.884_{\pm.006}$ | $.005_{\pm.003}$ | $.998_{\pm.001}$ | $2.08_{\pm4.68}$ | $.989_{\pm.103}$ | $.801_{\pm.003}$ | .002 | 32(124) |
| Matryoshka (UW) | $.830_{\pm.003}$ | $.000_{\pm.000}$ | $.999_{\pm.000}$ | $0.12_{\pm0.41}$ | $.998_{\pm.050}$ | $.853_{\pm.002}$ | .002 | 22(491) |

*Table 8.* **CLIP ViT-L/14 SAE comparison at expansion rate 32.** Analysis at maximum tested expansion rate on ImageNet-1k, completing the scaling study from Tables 6 and 7.

| Model | $L_0 \uparrow$ | FVU $\downarrow$ | CS $\uparrow$ | LP (KL) $\downarrow$ | LP (Acc) $\uparrow$ | CKNNA $\uparrow$ | DO $\downarrow$ | NDN $\downarrow$ |
|---|---|---|---|---|---|---|---|---|
| ReLU ($\lambda = 0.03$) | $.964_{\pm.004}$ | $.120_{\pm.029}$ | $.948_{\pm.007}$ | $36.1_{\pm60.2}$ | $.949_{\pm.221}$ | $.707_{\pm.005}$ | .003 | 0(0) |
| ReLU ($\lambda = 0.01$) | $.893_{\pm.006}$ | $.032_{\pm.005}$ | $.985_{\pm.002}$ | $7.65_{\pm12.77}$ | $.977_{\pm.150}$ | $.752_{\pm.008}$ | .002 | 0(0) |
| ReLU ($\lambda = 0.003$) | $.781_{\pm.007}$ | $.011_{\pm.002}$ | $.995_{\pm.001}$ | $2.06_{\pm3.40}$ | $.988_{\pm.111}$ | $.619_{\pm.007}$ | .002 | 0(0) |
| ReLU ($\lambda = 0.001$) | $.653_{\pm.005}$ | $.004_{\pm.001}$ | $.998_{\pm.000}$ | $0.77_{\pm1.25}$ | $.993_{\pm.085}$ | $.493_{\pm.007}$ | .002 | 0(0) |
| TopK ($k = 32$) | $.938_{\pm.018}$ | $.246_{\pm.047}$ | $.872_{\pm.025}$ | $102.37_{\pm155.09}$ | $.906_{\pm.292}$ | $.697_{\pm.012}$ | .002 | 64(14535) |
| TopK ($k = 64$) | $.973_{\pm.006}$ | $.174_{\pm.028}$ | $.911_{\pm.014}$ | $61.7_{\pm96.8}$ | $.927_{\pm.260}$ | $.745_{\pm.003}$ | .002 | 0(9347) |
| TopK ($k = 128$) | $.964_{\pm.009}$ | $.110_{\pm.021}$ | $.947_{\pm.009}$ | $31.4_{\pm51.2}$ | $.952_{\pm.213}$ | $.794_{\pm.005}$ | .002 | 10(5604) |
| TopK ($k = 256$) | $.942_{\pm.012}$ | $.032_{\pm.008}$ | $.986_{\pm.003}$ | $5.39_{\pm8.80}$ | $.980_{\pm.139}$ | $.864_{\pm.003}$ | .003 | 91(6590) |
| TopK ($k = 512$) | $.966_{\pm.004}$ | $.008_{\pm.005}$ | $.996_{\pm.002}$ | $1.30_{\pm2.69}$ | $.992_{\pm.092}$ | $.422_{\pm.018}$ | .002 | 7822(22446) |
| BatchTopK ($k = 16$) | $.638_{\pm.018}$ | $.462_{\pm.057}$ | $.775_{\pm.038}$ | $344.4_{\pm377.9}$ | $.825_{\pm.380}$ | $.735_{\pm.005}$ | .001 | 0(21631) |
| BatchTopK ($k = 32$) | $.756_{\pm.015}$ | $.329_{\pm.042}$ | $.855_{\pm.024}$ | $139.4_{\pm186.8}$ | $.894_{\pm.308}$ | $.747_{\pm.004}$ | .001 | 0(18965) |
| BatchTopK ($k = 64$) | $.880_{\pm.011}$ | $.190_{\pm.024}$ | $.911_{\pm.013}$ | $55.4_{\pm84.7}$ | $.934_{\pm.249}$ | $.764_{\pm.004}$ | .002 | 0(15035) |
| BatchTopK ($k = 128$) | $.869_{\pm.012}$ | $.129_{\pm.016}$ | $.947_{\pm.007}$ | $26.7_{\pm42.2}$ | $.956_{\pm.206}$ | $.794_{\pm.003}$ | .002 | 1(11019) |
| BatchTopK ($k = 256$) | $.837_{\pm.014}$ | $.069_{\pm.011}$ | $.979_{\pm.004}$ | $8.22_{\pm13.4}$ | $.976_{\pm.154}$ | $.851_{\pm.003}$ | .002 | 12(11802) |
| Matryoshka (RW) | $.927_{\pm.004}$ | $.003_{\pm.001}$ | $.999_{\pm.001}$ | $1.00_{\pm2.15}$ | $.992_{\pm.090}$ | $.810_{\pm.002}$ | .002 | 79(142) |
| Matryoshka (UW) | $.908_{\pm.002}$ | $.000_{\pm.000}$ | $.999_{\pm.000}$ | $0.09_{\pm0.35}$ | $.998_{\pm.047}$ | $.850_{\pm.003}$ | .002 | 297(162) |

*Table 9.* **CLIP ViT-B/16 SAE comparison at expansion rate 8.** Parallel analysis to ViT-L/14 (Table 6) using smaller CLIP architecture on ImageNet-1k.

| Model | $L_0 \uparrow$ | FVU $\downarrow$ | CS $\uparrow$ | CKNNA $\uparrow$ | DO $\downarrow$ | NDN $\downarrow$ |
|---|---|---|---|---|---|---|
| ReLU ($\lambda = 0.03$) | $.908_{\pm.010}$ | $.154_{\pm.036}$ | $.936_{\pm.010}$ | $.671_{\pm.004}$ | .004 | 0(0) |
| ReLU ($\lambda = 0.01$) | $.747_{\pm.012}$ | $.030_{\pm.005}$ | $.986_{\pm.002}$ | $.682_{\pm.005}$ | .003 | 0(0) |
| ReLU ($\lambda = 0.003$) | $.629_{\pm.008}$ | $.003_{\pm.000}$ | $.999_{\pm.000}$ | $.737_{\pm.003}$ | .003 | 0(0) |
| ReLU ($\lambda = 0.001$) | $.366_{\pm.012}$ | $.009_{\pm.003}$ | $.996_{\pm.001}$ | $.695_{\pm.003}$ | .002 | 0(0) |
| TopK ($k = 32$) | $.946_{\pm.013}$ | $.200_{\pm.037}$ | $.897_{\pm.019}$ | $.684_{\pm.005}$ | .004 | 0(672) |
| TopK ($k = 64$) | $.935_{\pm.011}$ | $.138_{\pm.024}$ | $.930_{\pm.012}$ | $.730_{\pm.003}$ | .003 | 0(196) |
| TopK ($k = 128$) | $.843_{\pm.020}$ | $.116_{\pm.026}$ | $.948_{\pm.010}$ | $.735_{\pm.003}$ | .003 | 0(95) |
| TopK ($k = 256$) | $.859_{\pm.008}$ | $.024_{\pm.007}$ | $.988_{\pm.003}$ | $.787_{\pm.002}$ | .004 | 0(8) |
| TopK ($k = 512$) | $.882_{\pm.008}$ | $.058_{\pm.052}$ | $.972_{\pm.025}$ | $.005_{\pm.003}$ | .003 | 0(0) |
| BatchTopK ($k = 16$) | $.706_{\pm.023}$ | $.299_{\pm.050}$ | $.845_{\pm.030}$ | $.662_{\pm.005}$ | .002 | 0(2666) |
| BatchTopK ($k = 32$) | $.810_{\pm.019}$ | $.196_{\pm.031}$ | $.900_{\pm.017}$ | $.695_{\pm.005}$ | .003 | 0(1763) |
| BatchTopK ($k = 64$) | $.877_{\pm.013}$ | $.127_{\pm.020}$ | $.936_{\pm.010}$ | $.750_{\pm.004}$ | .003 | 0(830) |
| BatchTopK ($k = 128$) | $.882_{\pm.010}$ | $.048_{\pm.008}$ | $.976_{\pm.004}$ | $.806_{\pm.003}$ | .003 | 0(387) |
| BatchTopK ($k = 256$) | $.876_{\pm.003}$ | $.003_{\pm.001}$ | $.999_{\pm.001}$ | $.843_{\pm.003}$ | .003 | 35(1766) |
| Matryoshka (RW) | $.783_{\pm.009}$ | $.003_{\pm.001}$ | $.999_{\pm.001}$ | $.792_{\pm.003}$ | .003 | 0(0) |
| Matryoshka (UW) | $.711_{\pm.004}$ | $.000_{\pm.000}$ | $.999_{\pm.000}$ | $.814_{\pm.002}$ | .003 | 0(1) |

*Table 10.* **CLIP ViT-B/16 SAE comparison at expansion rate 16.** Extension of Table 9 to expansion rate 16 on ImageNet-1k.

| Model | $L_0 \uparrow$ | FVU $\downarrow$ | CS $\uparrow$ | CKNNA $\uparrow$ | DO $\downarrow$ | NDN $\downarrow$ |
|---|---|---|---|---|---|---|
| ReLU ($\lambda = 0.03$) | $.940_{\pm.007}$ | $.125_{\pm.030}$ | $.945_{\pm.008}$ | $.664_{\pm.006}$ | .004 | 0(0) |
| ReLU ($\lambda = 0.01$) | $.824_{\pm.010}$ | $.033_{\pm.005}$ | $.984_{\pm.002}$ | $.669_{\pm.006}$ | .003 | 0(0) |
| ReLU ($\lambda = 0.003$) | $.695_{\pm.010}$ | $.004_{\pm.008}$ | $.998_{\pm.000}$ | $.635_{\pm.005}$ | .003 | 0(0) |
| ReLU ($\lambda = 0.001$) | $.646_{\pm.008}$ | $.001_{\pm.000}$ | $.999_{\pm.000}$ | $.742_{\pm.003}$ | .003 | 0(0) |
| TopK ($k = 32$) | $.958_{\pm.015}$ | $.206_{\pm.042}$ | $.896_{\pm.021}$ | $.671_{\pm.005}$ | .003 | 0(2819) |
| TopK ($k = 64$) | $.962_{\pm.008}$ | $.141_{\pm.026}$ | $.930_{\pm.012}$ | $.710_{\pm.003}$ | .003 | 0(1268) |
| TopK ($k = 128$) | $.950_{\pm.007}$ | $.072_{\pm.016}$ | $.965_{\pm.007}$ | $.782_{\pm.003}$ | .004 | 0(597) |
| TopK ($k = 256$) | $.935_{\pm.003}$ | $.003_{\pm.002}$ | $.998_{\pm.001}$ | $.839_{\pm.002}$ | .003 | 2(4686) |
| TopK ($k = 512$) | $.943_{\pm.015}$ | $.203_{\pm.312}$ | $.955_{\pm.047}$ | $.006_{\pm.003}$ | .003 | 1(1) |
| BatchTopK ($k = 16$) | $.658_{\pm.022}$ | $.349_{\pm.054}$ | $.837_{\pm.033}$ | $.670_{\pm.006}$ | .002 | 0(6478) |
| BatchTopK ($k = 32$) | $.757_{\pm.022}$ | $.230_{\pm.035}$ | $.896_{\pm.019}$ | $.704_{\pm.005}$ | .002 | 0(5114) |
| BatchTopK ($k = 64$) | $.853_{\pm.016}$ | $.135_{\pm.019}$ | $.938_{\pm.009}$ | $.741_{\pm.005}$ | .003 | 0(3145) |
| BatchTopK ($k = 128$) | $.887_{\pm.010}$ | $.061_{\pm.011}$ | $.971_{\pm.005}$ | $.799_{\pm.004}$ | .003 | 0(1817) |
| BatchTopK ($k = 256$) | $.876_{\pm.003}$ | $.003_{\pm.001}$ | $.999_{\pm.001}$ | $.843_{\pm.003}$ | .002 | 40(4947) |
| Matryoshka (RW) | $.861_{\pm.005}$ | $.002_{\pm.001}$ | $.999_{\pm.001}$ | $.778_{\pm.003}$ | .003 | 8(63) |
| Matryoshka (UW) | $.805_{\pm.004}$ | $.000_{\pm.000}$ | $.999_{\pm.000}$ | $.813_{\pm.003}$ | .003 | 44(275) |

*Table 11.* **CLIP ViT-B/16 SAE comparison at expansion rate 32.** Completion of ViT-B/16 scaling analysis on ImageNet-1k at maximum tested expansion rate.

| Model | $L_0 \uparrow$ | FVU $\downarrow$ | CS $\uparrow$ | CKNNA $\uparrow$ | DO $\downarrow$ | NDN $\downarrow$ |
|---|---|---|---|---|---|---|
| ReLU ($\lambda = 0.03$) | $.956_{\pm.005}$ | $.104_{\pm.025}$ | $.953_{\pm.007}$ | $.656_{\pm.004}$ | .003 | 0(0) |
| ReLU ($\lambda = 0.01$) | $.879_{\pm.008}$ | $.031_{\pm.005}$ | $.986_{\pm.002}$ | $.688_{\pm.005}$ | .003 | 0(0) |
| ReLU ($\lambda = 0.003$) | $.757_{\pm.009}$ | $.010_{\pm.002}$ | $.996_{\pm.001}$ | $.568_{\pm.005}$ | .003 | 0(0) |
| ReLU ($\lambda = 0.001$) | $.625_{\pm.006}$ | $.004_{\pm.001}$ | $.998_{\pm.000}$ | $.516_{\pm.005}$ | .003 | 0(0) |
| TopK ($k = 32$) | $.933_{\pm.025}$ | $.230_{\pm.058}$ | $.891_{\pm.025}$ | $.674_{\pm.005}$ | .003 | 0(9223) |
| TopK ($k = 64$) | $.967_{\pm.012}$ | $.152_{\pm.032}$ | $.927_{\pm.014}$ | $.698_{\pm.003}$ | .003 | 0(5643) |
| TopK ($k = 128$) | $.960_{\pm.010}$ | $.085_{\pm.023}$ | $.961_{\pm.009}$ | $.772_{\pm.002}$ | .003 | 2(3321) |
| TopK ($k = 256$) | $.922_{\pm.014}$ | $.015_{\pm.006}$ | $.995_{\pm.002}$ | $.822_{\pm.002}$ | .003 | 1(10480) |
| TopK ($k = 512$) | $.967_{\pm.000}$ | $.001_{\pm.001}$ | $1.000_{\pm.001}$ | $.016_{\pm.007}$ | .002 | 5902(14864) |
| BatchTopK ($k = 16$) | $.615_{\pm.018}$ | $.450_{\pm.060}$ | $.822_{\pm.037}$ | $.695_{\pm.005}$ | .002 | 0(14345) |
| BatchTopK ($k = 32$) | $.712_{\pm.021}$ | $.312_{\pm.043}$ | $.890_{\pm.019}$ | $.719_{\pm.005}$ | .002 | 0(12492) |
| BatchTopK ($k = 64$) | $.835_{\pm.017}$ | $.184_{\pm.028}$ | $.933_{\pm.010}$ | $.731_{\pm.005}$ | .002 | 2(9645) |
| BatchTopK ($k = 128$) | $.856_{\pm.013}$ | $.096_{\pm.015}$ | $.966_{\pm.005}$ | $.798_{\pm.003}$ | .003 | 0(6855) |
| BatchTopK ($k = 256$) | $.844_{\pm.011}$ | $.040_{\pm.007}$ | $.991_{\pm.002}$ | $.827_{\pm.003}$ | .002 | 0(11638) |
| Matryoshka (RW) | $.915_{\pm.003}$ | $.001_{\pm.001}$ | $.999_{\pm.000}$ | $.794_{\pm.003}$ | .003 | 6(23) |
| Matryoshka (UW) | $.880_{\pm.003}$ | $.000_{\pm.000}$ | $.999_{\pm.000}$ | $.804_{\pm.002}$ | .002 | 15(26) |

## F.5.2. RESULTS FOR TEXT MODALITY

For text modality, Tables 12–14 present results for ViT-L/14, while Tables 15–17 show ViT-B/16 performance on CC3M validation text data, enabling cross-modal and cross-architecture comparisons.

*Table 12.* **CLIP ViT-L/14 SAE text analysis at expansion rate 8.** Evaluation on CC3M text validation set parallel to image results in Table 6, highlighting cross-modal performance differences.

| Model | $L_0 \uparrow$ | FVU $\downarrow$ | CS $\uparrow$ | CKNNA $\uparrow$ | DO $\downarrow$ | NDN $\downarrow$ |
|---|---|---|---|---|---|---|
| ReLU ($\lambda = 0.03$) | $.901_{\pm.010}$ | $.427_{\pm.174}$ | $.802_{\pm.049}$ | $.622_{\pm.003}$ | .003 | 0(0) |
| ReLU ($\lambda = 0.01$) | $.522_{\pm.041}$ | $.036_{\pm.038}$ | $.984_{\pm.014}$ | $.716_{\pm.006}$ | .002 | 0(0) |
| ReLU ($\lambda = 0.003$) | $.609_{\pm.027}$ | $.041_{\pm.039}$ | $.981_{\pm.018}$ | $.744_{\pm.002}$ | .003 | 0(0) |
| ReLU ($\lambda = 0.001$) | $.522_{\pm.045}$ | $.035_{\pm.038}$ | $.984_{\pm.014}$ | $.706_{\pm.005}$ | .002 | 0(0) |
| TopK ($k = 32$) | $.724_{\pm.318}$ | $1.095_{\pm.835}$ | $.511_{\pm.206}$ | $.037_{\pm.031}$ | .002 | 0(1235) |
| TopK ($k = 64$) | $.715_{\pm.295}$ | $.760_{\pm.249}$ | $.585_{\pm.209}$ | $.025_{\pm.020}$ | .002 | 0(335) |
| TopK ($k = 128$) | $.781_{\pm.190}$ | $.537_{\pm.252}$ | $.708_{\pm.190}$ | $.042_{\pm.011}$ | .003 | 0(117) |
| TopK ($k = 256$) | $.783_{\pm.180}$ | $.366_{\pm.366}$ | $.742_{\pm.301}$ | $.088_{\pm.007}$ | .003 | 0(296) |
| TopK ($k = 512$) | $.900_{\pm.026}$ | $.386_{\pm.376}$ | $.865_{\pm.101}$ | $.038_{\pm.006}$ | .002 | 0(1) |
| BatchTopK ($k = 16$) | $.585_{\pm.161}$ | $1.222_{\pm.929}$ | $.416_{\pm.213}$ | $.031_{\pm.026}$ | .002 | 0(4278) |
| BatchTopK ($k = 32$) | $.608_{\pm.221}$ | $.848_{\pm.277}$ | $.494_{\pm.232}$ | $.022_{\pm.012}$ | .002 | 0(3080) |
| BatchTopK ($k = 64$) | $.712_{\pm.197}$ | $.662_{\pm.229}$ | $.581_{\pm.222}$ | $.019_{\pm.013}$ | .002 | 0(1477) |
| BatchTopK ($k = 128$) | $.858_{\pm.038}$ | $.415_{\pm.180}$ | $.787_{\pm.102}$ | $.312_{\pm.009}$ | .003 | 0(539) |
| BatchTopK ($k = 256$) | $.869_{\pm.026}$ | $.159_{\pm.175}$ | $.918_{\pm.108}$ | $.716_{\pm.004}$ | .002 | 17(919) |
| Matryoshka (RW) | $.824_{\pm.029}$ | $.060_{\pm.052}$ | $.971_{\pm.026}$ | $.775_{\pm.001}$ | .002 | 0(4) |
| Matryoshka (UW) | $.755_{\pm.024}$ | $.026_{\pm.027}$ | $.988_{\pm.012}$ | $.790_{\pm.002}$ | .001 | 0(22) |

*Table 13.* **CLIP ViT-L/14 SAE text analysis at expansion rate 16.** Extended CC3M text evaluation showing scaling effects at expansion rate 16, complementing image results from Table 7.

| Model | $L_0 \uparrow$ | FVU $\downarrow$ | CS $\uparrow$ | CKNNA $\uparrow$ | DO $\downarrow$ | NDN $\downarrow$ |
|---|---|---|---|---|---|---|
| ReLU ($\lambda = 0.03$) | $.930_{\pm.027}$ | $.510_{\pm.500}$ | $.812_{\pm.052}$ | $.581_{\pm.006}$ | .003 | 0(0) |
| ReLU ($\lambda = 0.01$) | $.809_{\pm.052}$ | $.221_{\pm.255}$ | $.926_{\pm.043}$ | $.643_{\pm.008}$ | .002 | 0(0) |
| ReLU ($\lambda = 0.003$) | $.675_{\pm.067}$ | $.070_{\pm.060}$ | $.973_{\pm.023}$ | $.654_{\pm.009}$ | .002 | 0(0) |
| ReLU ($\lambda = 0.001$) | $.599_{\pm.028}$ | $.021_{\pm.019}$ | $.990_{\pm.010}$ | $.781_{\pm.002}$ | .002 | 0(0) |
| TopK ($k = 32$) | $.782_{\pm.283}$ | $1.237_{\pm1.122}$ | $.494_{\pm.208}$ | $.038_{\pm.028}$ | .002 | 0(4727) |
| TopK ($k = 64$) | $.774_{\pm.280}$ | $.790_{\pm.258}$ | $.583_{\pm.203}$ | $.023_{\pm.008}$ | .002 | 0(2079) |
| TopK ($k = 128$) | $.813_{\pm.212}$ | $.604_{\pm.255}$ | $.690_{\pm.190}$ | $.029_{\pm.010}$ | .002 | 0(897) |
| TopK ($k = 256$) | $.848_{\pm.156}$ | $.390_{\pm.357}$ | $.740_{\pm.280}$ | $.093_{\pm.004}$ | .004 | 0(1383) |
| TopK ($k = 512$) | $.950_{\pm.023}$ | $.435_{\pm.453}$ | $.855_{\pm.107}$ | $.073_{\pm.014}$ | .002 | 0(29) |
| BatchTopK ($k = 16$) | $.600_{\pm.115}$ | $.917_{\pm.326}$ | $.450_{\pm.202}$ | $.032_{\pm.018}$ | .001 | 0(9859) |
| BatchTopK ($k = 32$) | $.619_{\pm.181}$ | $.758_{\pm.209}$ | $.518_{\pm.232}$ | $.031_{\pm.021}$ | .002 | 0(8016) |
| BatchTopK ($k = 64$) | $.706_{\pm.242}$ | $.687_{\pm.236}$ | $.559_{\pm.240}$ | $.029_{\pm.020}$ | .002 | 0(5113) |
| BatchTopK ($k = 128$) | $.866_{\pm.034}$ | $.465_{\pm.230}$ | $.796_{\pm.081}$ | $.240_{\pm.016}$ | .002 | 14(2967) |
| BatchTopK ($k = 256$) | $.789_{\pm.189}$ | $.352_{\pm.371}$ | $.741_{\pm.334}$ | $.036_{\pm.004}$ | .002 | 7(3558) |
| Matryoshka (RW) | $.880_{\pm.021}$ | $.043_{\pm.038}$ | $.980_{\pm.019}$ | $.783_{\pm.006}$ | .002 | 4(124) |
| Matryoshka (UW) | $.832_{\pm.028}$ | $.017_{\pm.017}$ | $.992_{\pm.008}$ | $.788_{\pm.001}$ | .002 | 0(491) |

*Table 14.* **CLIP ViT-L/14 SAE text analysis at expansion rate 32.** Maximum expansion rate analysis on CC3M text.

| Model | $L_0 \uparrow$ | FVU $\downarrow$ | CS $\uparrow$ | CKNNA $\uparrow$ | DO $\downarrow$ | NDN $\downarrow$ |
|---|---|---|---|---|---|---|
| ReLU ($\lambda = 0.03$) | $.951_{\pm.023}$ | $.540_{\pm.667}$ | $.822_{\pm.053}$ | $.557_{\pm.001}$ | .003 | 1(0) |
| ReLU ($\lambda = 0.01$) | $.867_{\pm.059}$ | $.339_{\pm.546}$ | $.927_{\pm.047}$ | $.549_{\pm.011}$ | .002 | 1(0) |
| ReLU ($\lambda = 0.003$) | $.749_{\pm.090}$ | $.172_{\pm.200}$ | $.966_{\pm.027}$ | $.336_{\pm.008}$ | .002 | 0(0) |
| ReLU ($\lambda = 0.001$) | $.631_{\pm.054}$ | $.052_{\pm.045}$ | $.983_{\pm.014}$ | $.376_{\pm.007}$ | .002 | 0(0) |
| TopK ($k = 32$) | $.864_{\pm.144}$ | $1.149_{\pm1.282}$ | $.551_{\pm.169}$ | $.058_{\pm.018}$ | .002 | 0(14535) |
| TopK ($k = 64$) | $.888_{\pm.156}$ | $.795_{\pm.337}$ | $.630_{\pm.157}$ | $.049_{\pm.026}$ | .002 | 0(2079) |
| TopK ($k = 128$) | $.871_{\pm.156}$ | $.612_{\pm.252}$ | $.717_{\pm.161}$ | $.053_{\pm.024}$ | .002 | 0(5604) |
| TopK ($k = 256$) | $.869_{\pm.134}$ | $.435_{\pm.342}$ | $.740_{\pm.240}$ | $.130_{\pm.006}$ | .003 | 0(6590) |
| TopK ($k = 512$) | $.937_{\pm.062}$ | $.115_{\pm.131}$ | $.944_{\pm.069}$ | $.070_{\pm.034}$ | .002 | 240(22446) |
| BatchTopK ($k = 16$) | $.601_{\pm.079}$ | $.818_{\pm.209}$ | $.496_{\pm.191}$ | $.034_{\pm.024}$ | .002 | 0(21631) |
| BatchTopK ($k = 32$) | $.681_{\pm.143}$ | $.733_{\pm.235}$ | $.592_{\pm.190}$ | $.034_{\pm.020}$ | .002 | 0(21631) |
| BatchTopK ($k = 64$) | $.784_{\pm.178}$ | $.636_{\pm.225}$ | $.657_{\pm.181}$ | $.042_{\pm.026}$ | .002 | 0(18965) |
| BatchTopK ($k = 128$) | $.861_{\pm.016}$ | $.377_{\pm.159}$ | $.830_{\pm.055}$ | $.526_{\pm.012}$ | .002 | 94(11019) |
| BatchTopK ($k = 256$) | $.784_{\pm.139}$ | $.332_{\pm.333}$ | $.778_{\pm.288}$ | $.060_{\pm.007}$ | .002 | 0(11802) |
| Matryoshka (RW) | $.925_{\pm.014}$ | $.030_{\pm.026}$ | $.986_{\pm.013}$ | $.774_{\pm.000}$ | .002 | 32(142) |
| Matryoshka (UW) | $.901_{\pm.026}$ | $.013_{\pm.013}$ | $.994_{\pm.006}$ | $.784_{\pm.000}$ | .002 | 126(162) |

*Table 15.* **CLIP ViT-B/16 SAE text analysis at expansion rate 8.** CC3M text evaluation using smaller CLIP architecture, enabling cross-modal and cross-architecture comparisons.

| Model | $L_0 \uparrow$ | FVU $\downarrow$ | CS $\uparrow$ | CKNNA $\uparrow$ | DO $\downarrow$ | NDN $\downarrow$ |
|---|---|---|---|---|---|---|
| ReLU ($\lambda = 0.03$) | $.870_{\pm.021}$ | $.472_{\pm.166}$ | $.761_{\pm.063}$ | $.661_{\pm.008}$ | .004 | 0(0) |
| ReLU ($\lambda = 0.01$) | $.697_{\pm.046}$ | $.183_{\pm.141}$ | $.920_{\pm.048}$ | $.721_{\pm.003}$ | .003 | 0(0) |
| ReLU ($\lambda = 0.003$) | $.580_{\pm.028}$ | $.030_{\pm.028}$ | $.986_{\pm.013}$ | $.764_{\pm.000}$ | .003 | 0(0) |
| ReLU ($\lambda = 0.001$) | $.393_{\pm.122}$ | $.118_{\pm.177}$ | $.975_{\pm.022}$ | $.129_{\pm.021}$ | .002 | 0(0) |
| TopK ($k = 32$) | $.757_{\pm.265}$ | $1.095_{\pm1.081}$ | $.533_{\pm.174}$ | $.035_{\pm.013}$ | .004 | 0(672) |
| TopK ($k = 64$) | $.766_{\pm.223}$ | $.733_{\pm.353}$ | $.644_{\pm.155}$ | $.033_{\pm.001}$ | .003 | 0(196) |
| TopK ($k = 128$) | $.747_{\pm.164}$ | $.515_{\pm.343}$ | $.782_{\pm.106}$ | $.275_{\pm.016}$ | .003 | 0(95) |
| TopK ($k = 256$) | $.783_{\pm.095}$ | $.229_{\pm.152}$ | $.888_{\pm.081}$ | $.759_{\pm.000}$ | .004 | 0(8) |
| TopK ($k = 512$) | $.794_{\pm.174}$ | $.283_{\pm.282}$ | $.860_{\pm.154}$ | $.079_{\pm.021}$ | .003 | 0(0) |
| BatchTopK ($k = 16$) | $.602_{\pm.156}$ | $1.172_{\pm1.110}$ | $.461_{\pm.178}$ | $.018_{\pm.010}$ | .002 | 0(2666) |
| BatchTopK ($k = 32$) | $.662_{\pm.213}$ | $.898_{\pm.510}$ | $.547_{\pm.176}$ | $.016_{\pm.008}$ | .003 | 0(1763) |
| BatchTopK ($k = 64$) | $.716_{\pm.204}$ | $.656_{\pm.212}$ | $.637_{\pm.166}$ | $.029_{\pm.008}$ | .003 | 0(830) |
| BatchTopK ($k = 128$) | $.715_{\pm.270}$ | $.654_{\pm.322}$ | $.662_{\pm.256}$ | $.033_{\pm.009}$ | .003 | 0(387) |
| BatchTopK ($k = 256$) | $.774_{\pm.177}$ | $.317_{\pm.358}$ | $.776_{\pm.301}$ | $.103_{\pm.004}$ | .003 | 0(1766) |
| Matryoshka (RW) | $.762_{\pm.063}$ | $.044_{\pm.047}$ | $.979_{\pm.023}$ | $.799_{\pm.001}$ | .003 | 0(0) |
| Matryoshka (UW) | $.709_{\pm.043}$ | $.021_{\pm.025}$ | $.990_{\pm.012}$ | $.812_{\pm.003}$ | .003 | 0(1) |

*Table 16.* **CLIP ViT-B/16 SAE text analysis at expansion rate 16.** Results for expansion rate 16 with ViT-B/16 on CC3M text data.

| Model | $L_0 \uparrow$ | FVU $\downarrow$ | CS $\uparrow$ | CKNNA $\uparrow$ | DO $\downarrow$ | NDN $\downarrow$ |
|---|---|---|---|---|---|---|
| ReLU ($\lambda = 0.03$) | $.910_{\pm.023}$ | $.457_{\pm.178}$ | $.767_{\pm.067}$ | $.614_{\pm.008}$ | .004 | 0(0) |
| ReLU ($\lambda = 0.01$) | $.777_{\pm.065}$ | $.248_{\pm.244}$ | $.911_{\pm.049}$ | $.600_{\pm.003}$ | .003 | 0(0) |
| ReLU ($\lambda = 0.003$) | $.641_{\pm.055}$ | $.052_{\pm.038}$ | $.994_{\pm.005}$ | $.767_{\pm.001}$ | .003 | 0(0) |
| ReLU ($\lambda = 0.001$) | $.577_{\pm.037}$ | $.013_{\pm.012}$ | $.979_{\pm.016}$ | $.678_{\pm.002}$ | .003 | 0(0) |
| TopK ($k = 32$) | $.811_{\pm.232}$ | $1.163_{\pm1.376}$ | $.518_{\pm.185}$ | $.043_{\pm.020}$ | .003 | 0(2819) |
| TopK ($k = 64$) | $.801_{\pm.231}$ | $.803_{\pm.424}$ | $.618_{\pm.170}$ | $.028_{\pm.006}$ | .003 | 0(1268) |
| TopK ($k = 128$) | $.781_{\pm.217}$ | $.601_{\pm.252}$ | $.696_{\pm.177}$ | $.045_{\pm.002}$ | .004 | 0(597) |
| TopK ($k = 256$) | $.787_{\pm.224}$ | $.430_{\pm.401}$ | $.688_{\pm.326}$ | $.098_{\pm.007}$ | .003 | 0(4686) |
| TopK ($k = 512$) | $.918_{\pm.051}$ | $.447_{\pm.804}$ | $.891_{\pm.089}$ | $.070_{\pm.009}$ | .003 | 0(1) |
| BatchTopK ($k = 16$) | $.592_{\pm.109}$ | $.940_{\pm.510}$ | $.480_{\pm.187}$ | $.018_{\pm.007}$ | .002 | 0(6478) |
| BatchTopK ($k = 32$) | $.650_{\pm.174}$ | $.843_{\pm.428}$ | $.560_{\pm.182}$ | $.015_{\pm.007}$ | .002 | 0(5114) |
| BatchTopK ($k = 64$) | $.734_{\pm.174}$ | $.649_{\pm.212}$ | $.654_{\pm.153}$ | $.022_{\pm.006}$ | .003 | 0(3145) |
| BatchTopK ($k = 128$) | $.764_{\pm.218}$ | $.643_{\pm.310}$ | $.685_{\pm.216}$ | $.032_{\pm.004}$ | .003 | 0(1817) |
| BatchTopK ($k = 256$) | $.799_{\pm.194}$ | $.337_{\pm.375}$ | $.755_{\pm.321}$ | $.061_{\pm.006}$ | .002 | 0(4947) |
| Matryoshka (RW) | $.847_{\pm.040}$ | $.033_{\pm.036}$ | $.984_{\pm.018}$ | $.800_{\pm.002}$ | .003 | 0(63) |
| Matryoshka (UW) | $.801_{\pm.043}$ | $.017_{\pm.021}$ | $.992_{\pm.010}$ | $.803_{\pm.001}$ | .003 | 0(275) |

*Table 17.* **CLIP ViT-B/16 SAE text analysis at expansion rate 32.** Final expansion rate evaluation for ViT-B/16 on CC3M text.

| Model | $L_0 \uparrow$ | FVU $\downarrow$ | CS $\uparrow$ | CKNNA $\uparrow$ | DO $\downarrow$ | NDN $\downarrow$ |
|---|---|---|---|---|---|---|
| ReLU ($\lambda = 0.03$) | $.934_{\pm.019}$ | $.472_{\pm.356}$ | $.790_{\pm.058}$ | $.610_{\pm.008}$ | .003 | 0(0) |
| ReLU ($\lambda = 0.01$) | $.828_{\pm.123}$ | $.839_{\pm1.624}$ | $.894_{\pm.077}$ | $.145_{\pm.015}$ | .003 | 0(0) |
| ReLU ($\lambda = 0.003$) | $.713_{\pm.103}$ | $.183_{\pm.201}$ | $.963_{\pm.028}$ | $.304_{\pm.004}$ | .003 | 0(0) |
| ReLU ($\lambda = 0.001$) | $.600_{\pm.051}$ | $.041_{\pm.033}$ | $.987_{\pm.009}$ | $.422_{\pm.002}$ | .003 | 0(0) |
| TopK ($k = 32$) | $.850_{\pm.158}$ | $1.091_{\pm1.367}$ | $.547_{\pm.163}$ | $.076_{\pm.008}$ | .003 | 0(9223) |
| TopK ($k = 64$) | $.879_{\pm.146}$ | $.832_{\pm.622}$ | $.645_{\pm.138}$ | $.075_{\pm.028}$ | .003 | 0(5643) |
| TopK ($k = 128$) | $.863_{\pm.145}$ | $.590_{\pm.247}$ | $.740_{\pm.126}$ | $.150_{\pm.029}$ | .003 | 0(3321) |
| TopK ($k = 256$) | $.793_{\pm.215}$ | $.365_{\pm.359}$ | $.776_{\pm.253}$ | $.197_{\pm.024}$ | .003 | 0(10480) |
| TopK ($k = 512$) | $.917_{\pm.109}$ | $.097_{\pm.172}$ | $.955_{\pm.087}$ | $.027_{\pm.016}$ | .002 | 73(14864) |
| BatchTopK ($k = 16$) | $.581_{\pm.066}$ | $.769_{\pm.191}$ | $.536_{\pm.157}$ | $.027_{\pm.012}$ | .002 | 0(14345) |
| BatchTopK ($k = 32$) | $.657_{\pm.113}$ | $.726_{\pm.284}$ | $.628_{\pm.136}$ | $.032_{\pm.020}$ | .002 | 0(12492) |
| BatchTopK ($k = 64$) | $.766_{\pm.126}$ | $.631_{\pm.227}$ | $.704_{\pm.113}$ | $.052_{\pm.024}$ | .002 | 0(9645) |
| BatchTopK ($k = 128$) | $.783_{\pm.163}$ | $.628_{\pm.335}$ | $.727_{\pm.166}$ | $.046_{\pm.016}$ | .003 | 0(6855) |
| BatchTopK ($k = 256$) | $.773_{\pm.172}$ | $.311_{\pm.339}$ | $.799_{\pm.270}$ | $.117_{\pm.006}$ | .002 | 0(11638) |
| Matryoshka (RW) | $.897_{\pm.035}$ | $.022_{\pm.026}$ | $.990_{\pm.013}$ | $.807_{\pm.002}$ | .003 | 1(23) |
| Matryoshka (UW) | $.873_{\pm.030}$ | $.010_{\pm.014}$ | $.995_{\pm.007}$ | $.806_{\pm.003}$ | .002 | 0(26) |

# G. Interpreting CLIP with MSAE: Additional Results

In this appendix section, we provide additional analysis supporting Section 5. Section G.1 presents high-magnitude activation samples across modalities from MSAE (RW) with an expansion rate of 8. Section G.2 demonstrates how SAE enhances similarity search with interpretable results. Section G.3 presents statistical gender bias analysis on CelebA dataset, supported by concept manipulation visualizations that reinforce the statistical findings. These analyses strengthen our findings from Section 5 while providing deeper insights into MSAE's interpretability capabilities.

## G.1. Concept Visualization Analysis

Figures 18 and 19 showcase six valid concepts through their highest-activating images and texts, confirming concept validity. Conversely, Figure 20 demonstrates two invalid concepts, highlighting the importance of validation methods from Section A.

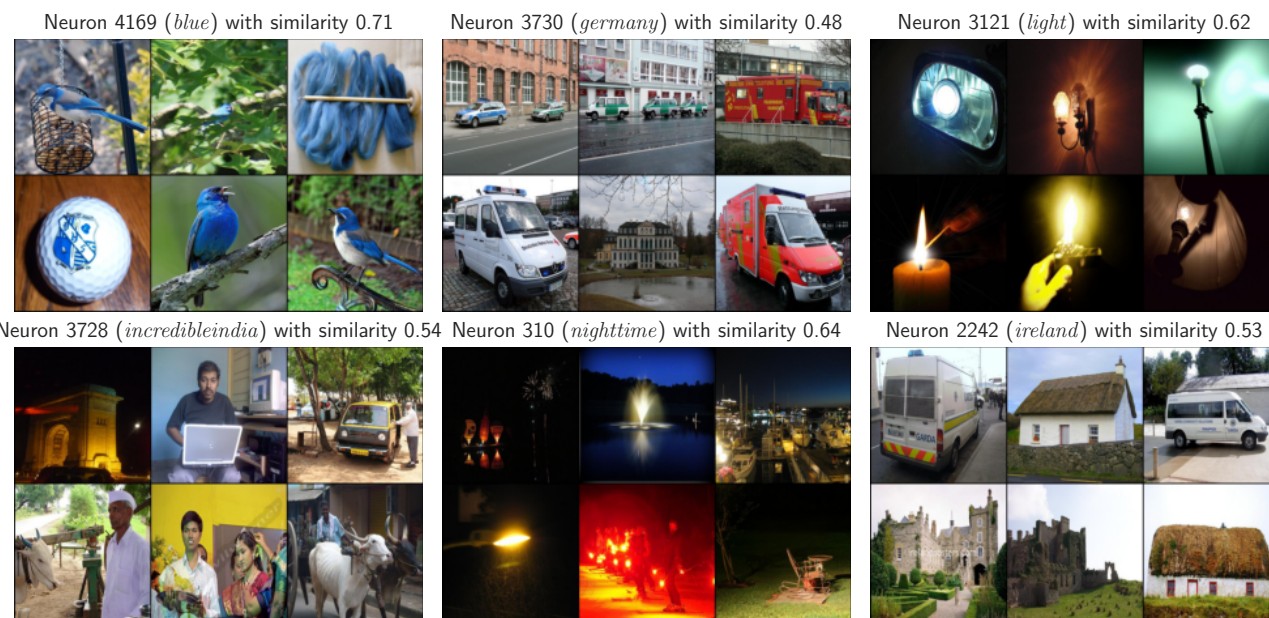

*Figure 18.* **High-magnitude image activations for valid concepts.** We gather top activating ImageNet-1k images for six valid MSAE (RW) concept neurons.

Neuron 3552 (*smile*) with similarity 0.59

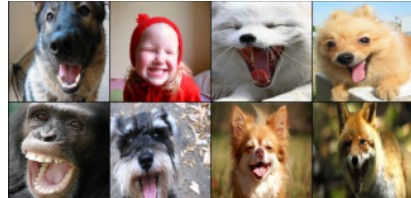

1. because this smile can cure the blues .
2. person with a lovely smile
3. portrait of a smiling boy
4. portrait of a senior man smiling
5. detail view of a pretty middle aged woman smiling
6. even after lots of walking ... all smiles !
7. man smiling to the camera

(a)

Neuron 3091 (*alcoholic*) with similarity 0.52

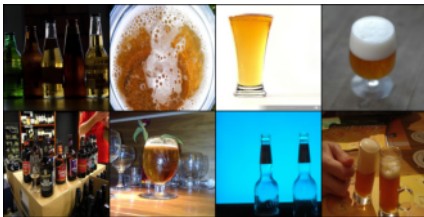

1. beverage type - the original beer
2. pouring beer into a glass , over a green background
3. customers drink beer at a bar
4. set of full beer bottles with no labels isolated on white
5. alcohol empty bottles in a window after a party
6. young woman with a mug of beer

(b)

Neuron 5071 (*trio*) with similarity 0.61

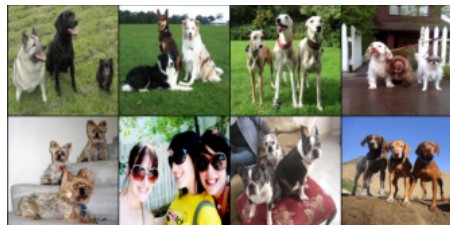

1. person , person and myself before the race
2. myself and the newlyweds at wedding that person at a private residence
3. putting a brave face on : the couple with person shortly before they announced their split
4. family of three on summer vacation .
5. backstage beauties : person happily posed with person and pop artist after the main event

(c)

Neuron 2539 (*heart*) with similarity 0.50

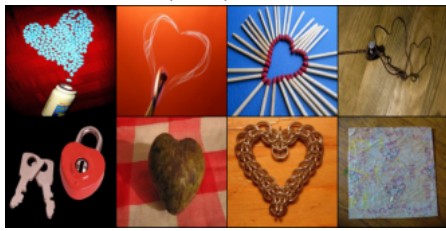

1. a symbolic electronic circuit in heart shape
2. illustration of the flag shaped like a heart .
3. heart shape single wine holder capable of holding wine glasses .
4. love is in the air
5. this heart was waiting for us as we climbed the stairs
6. flag with a pole and on a heart shape

(d)

Neuron 220 (*runnin*) with similarity 0.55

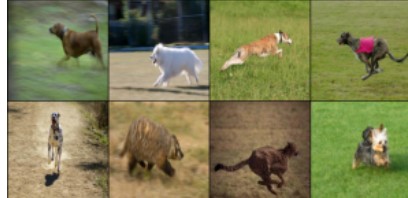

1. sisters running to the beach
2. person running at the beach
3. friends running towards the sea at the beach
4. couple running from the sea
5. teenage soccer players running down the field
6. person , is seen running in the streets .
7. person runs through the wilderness .

(e)

Neuron 2495 (*questions*) with similarity 0.53

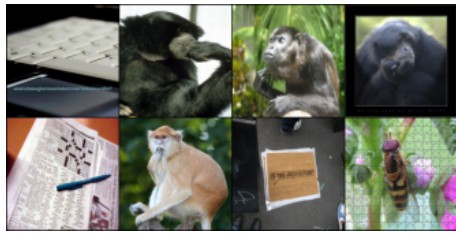

1. person asks questions monday of people arriving at the emergency room .
2. which players could leave football team in january ?
3. animal wants to know if you would like to hold it next ?
4. in what year did the bridge undergo an award - winning refurbishment ?
5. name the team to face a city

(f)

*Figure 19.* **Cross-modal highest valid concept activation samples.** Extending Figure 8, we show the highest-activating ImageNet-1k images and CC3M texts from valid MSAE (RW) concepts: *smile*, *alcoholic*, *trio*, *heart*, *running*, and *questions*.

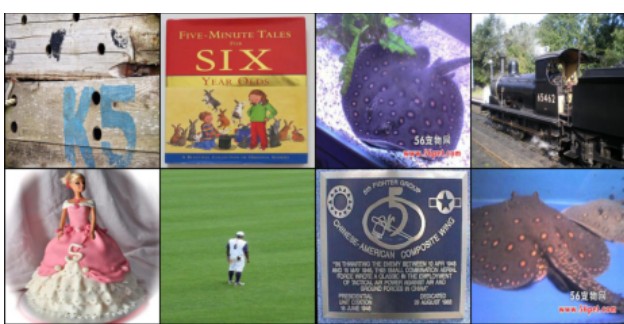

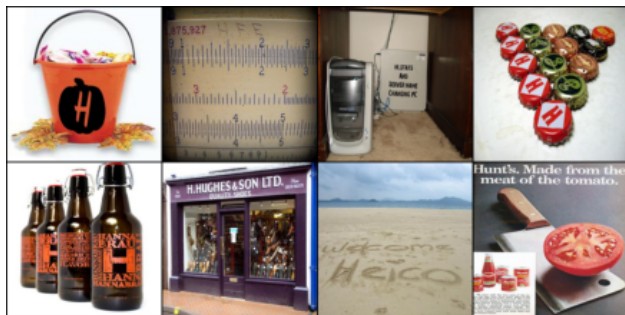

Concept 6 with two highest neuron similarities 0.4 and 0.17

Texts:

1. where sediment comes from 6 .

2. blues artist performs on stage during the fifth day

3. 60th anniversary : a sneak peek at new food and drinks

4. smile during the photo call for her new film at the 58th

5. students participate in activities for the 75th anniversary celebrations .

6. reserve at sixty three 63 has a wide variety of spacious floor plans

7. concert poster for folk rock artist and the 50th anniversary of composition .

8. the numbers and two painted on a dilapidated brick wall

Neuron 17006 most active for: $hl$ (0.45) $hri$ (0.44) $hig$ (0.43)

Texts:

1. hard rock artist , the heavy band , performs a live concert

2. the hawk first entered service both as an advanced flying - training aircraft and a weapons

3. habitat the type of environment in which an organism lives ; where an organism is commonly found .

4. holiday : has a heritage that stems .

5. the harbour on a summer day

6. hand drawn head of hen , label on a white background .

7. the chocolate bar that was launched and is based on hill

8. put some different wheels on the hawk

(a)                                                                 (b)

*Figure 20.* **Analysis of invalid concept neurons in MSAE (RW).** In (a), we showcase the invalid concept '6' with a low similarity score ($< 0.42$), which shows inconsistent presence of the number six in the top active samples. In (b), we present how a low ratio threshold ($0.45/0.44 < 2$) can indicate a broader 'h' concept rather than a specific 'hl'/'hri' from the vocabulary.

## G.2. SAE-Enhanced Similarity Search

Building upon Section 5, we demonstrate how SAE enhances nearest neighbor (NN) search by revealing shared semantic concepts between query and retrieved images. Figure 21 illustrates how SAE uncovers interpretable features that drive CLIP's similarity assessments. Furthermore, we show that conducting similarity search directly in the SAE activation space produces comparable results to CLIP-based search while providing more semantically meaningful matches.

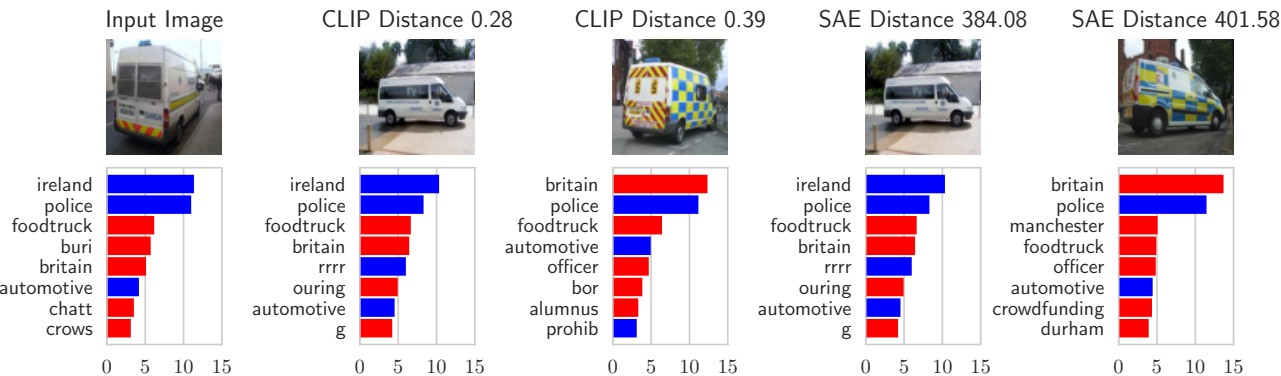

*Figure 21.* **SAE-enhanced similarity search.** Examples demonstrating how SAE reveals shared semantic concepts (bottom row) between query images and their CLIP nearest neighbors (top row), providing interpretable explanations for similarity matches. Additionally, the two rightmost examples show nearest neighbors retrieved based on SAE activation similarity, demonstrating how searching in the SAE space yields similar results to CLIP-based search while making the retrieval process more semantically interpretable.

## G.3. Gender Bias Analysis in CelebA

We analyze gender biases in a CLIP-based classification model using the CelebA dataset, which forms the foundation for our analysis in Section 5.3. Through statistical analysis of concept magnitude distribution against the model gender predictions in Figure 24, we identify significant gender associations for concepts *bearded*, *blondes*, and *glasses* in the classification model. To verify that these concepts align with the true features in the CelebA dataset, we visualize highest-activation images for each concept in Figure 22. Further concept manipulation experiments on both female (Figure 7) and male (Figure 23) examples confirm and strengthen these statistical findings, providing even greater insight into the relationship between gender classification and the chosen concepts.

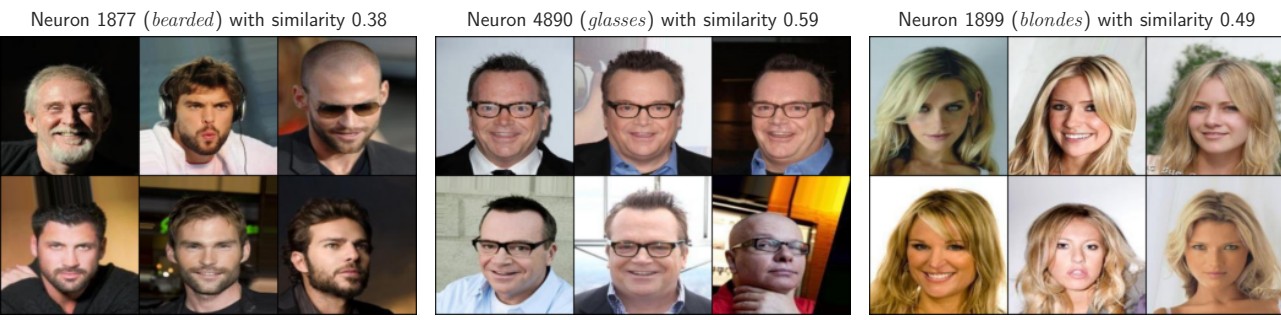

*Figure 22.* **Highest-activating CelebA images for gender-associated concepts.** We visualize images from the CelebA test set that produce the highest activations for the concepts *bearded*, *blondes*, and *glasses*, validating their alignment with the concept.

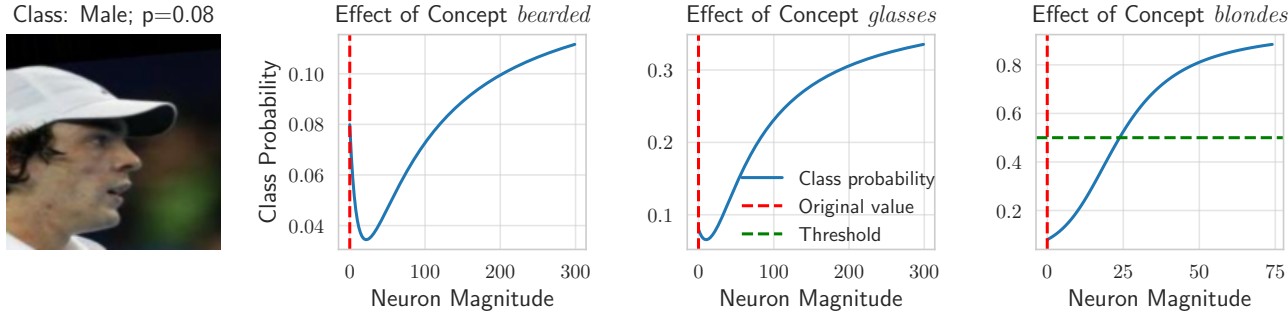

*Figure 23.* **Impact of concept manipulation for the male example.** Complementing Figure 7, we further strengthen our findings of male association for *bearded*, moderate for *glasses*, and female bias for *blondes* concept.

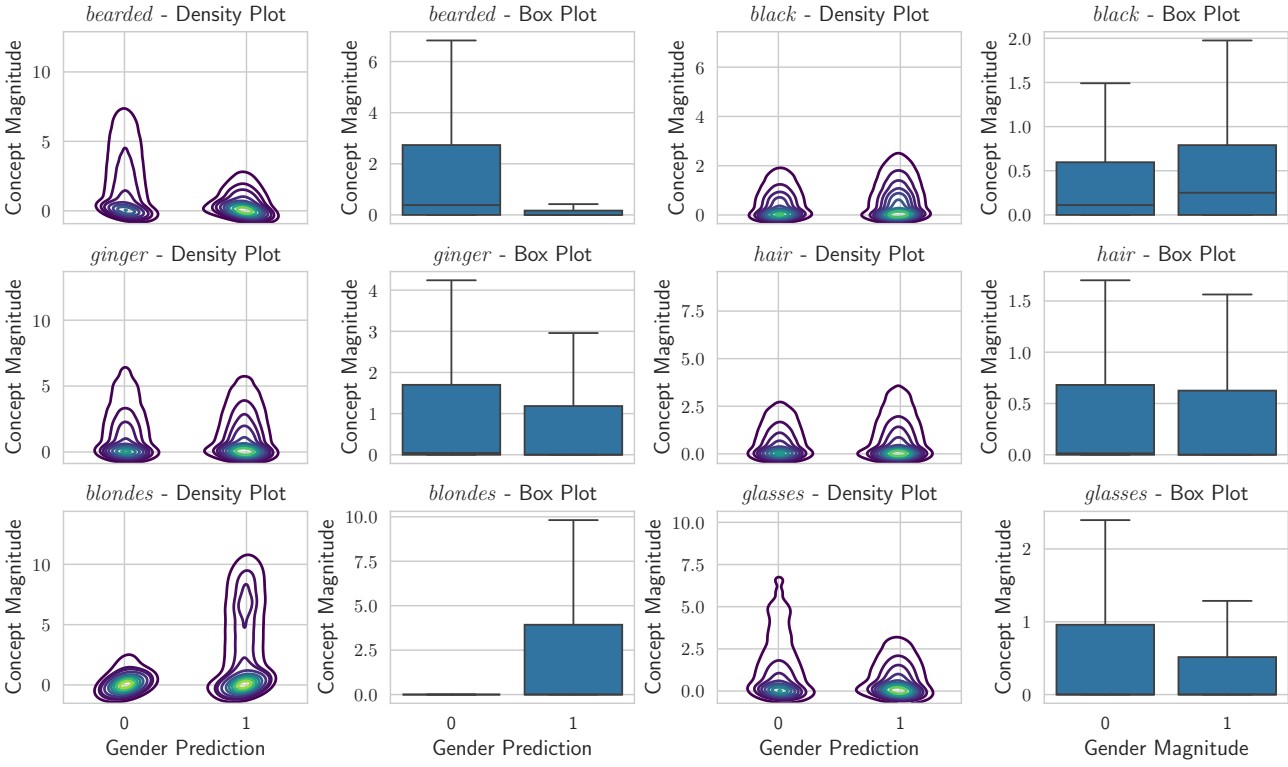

*Figure 24.* **Statistical analysis of concept-gender associations.** We analyze six concepts: *bearded*, *blondes*, *black*, *hair*, *glasses*, and *ginger*. For each concept, we show its density distribution of concept magnitude against gender prediction alongside corresponding boxplots. Results reveal that *bearded*, *blondes*, and *glasses* exhibit significant gender-specific associations.

# H. Stability Evaluations

To gain a deeper understanding of the stability of learned feature directions for the decoder and encoder across various training seeds, we calculated the stability metric proposed by Paulo & Belrose (2025). Table 18 shows the results for all tested SAEs at an expansion rate of 8. We observe that the stability metric is highly correlated with sparsity. Furthermore, Matryoshka SAE demonstrates a comparable stability trade-off to alternative architectures.

*Table 18.* **Stability–Sparsity–Reconstruction trade-off (Pareto front) for CLIP (ViT-L/14) on ImageNet-1k.** Rows are sorted by sparsity. We observe that (1) stability is highly correlated with sparsity, and (2) Matryoshka SAE exhibits an on-par stability trade-off as compared to other architectures.

| Model | Sparsity (L0 ↑) | Reconstruction (FVU ↓) | Stability (Decoder/Encoder ↑) |
|---|---|---|---|
| TopK ($k = 32$) | .960 | .245 | .649/.245 |
| TopK ($k = 64$) | .950 | .172 | .688/.240 |
| TopK ($k = 128$) | .928 | .098 | .625/.187 |
| TopK ($k = 512$) | .922 | .336 | .248/.187 |
| ReLU ($\lambda = 0.03$) | .920 | .185 | .522/.124 |
| TopK ($k = 256$) | .900 | .011 | .624/.235 |
| BatchTopK ($k = 128$) | .898 | .082 | .622/.238 |
| BatchTopK ($k = 256$) | .882 | .010 | .573/.231 |
| BatchTopK ($k = 64$) | .877 | .162 | .586/.238 |
| Matryoshka (RW) | .829 | .007 | .437/.102 |
| BatchTopK ($k = 32$) | .776 | .242 | .467/.168 |
| ReLU ($\lambda = 0.01$) | .762 | .033 | .401/.068 |
| Matryoshka (UW) | .748 | .002 | .366/.065 |
| BatchTopK ($k = 16$) | .698 | .371 | .352/.108 |
| ReLU ($\lambda = 0.003$) | .649 | .004 | .334/.042 |
| ReLU ($\lambda = 0.001$) | .553 | .002 | .200/.041 |

