# OpenReview forum: "Interpreting CLIP with Hierarchical Sparse Autoencoders"
_ICML.cc/2025/Conference — ICML 2025 poster_

### Official Review · Reviewer_T2kP · 2025-03-10

**Overall Recommendation:** 3

**Summary:**

The authors introduce Matryoshka SAE (MSAE), a novel Sparse Autoencoder (SAE) architecture that simultaneously learns hierarchical representations at multiple granularities. This is achieved by applying the topK operation multiple times while incrementally increasing the number of considered neurons. The proposed architecture exhibits an improved balance between reconstruction accuracy and sparsity compared to traditional SAEs (topK and ReLU SAEs). The authors apply MSAE to interpret CLIP embeddings and extract semantic concepts.

**Claims And Evidence:**

I have identified two main claims in this article :
1) MSAE achieves superior trade-offs between reconstruction and sparsity compared to traditional SAEs. This claim is backed by empirical results showing a better trade-off for MSAE against topK and ReLU SAEs.

2) MSAE effectively captures semantic concepts from CLIP embeddings. The authors propose various quantitative and qualitative analyses (concept naming, similarity search, bias validation).

**Essential References Not Discussed:**

I did not find any important reference missing

**Experimental Designs Or Analyses:**

Here are the weaknesses I found in the experiments :
1) Not enough data points (on topK and ReLU SAE) to strongly conclude that the MSAE exhibits a better reconstruction/sparsity than other SAEs (already mentioned before)
2) It would have been interesting to compare the MSAE to the jump ReLU SAE [1], as it is known to have a good reconstruction/sparsity tradeoff
3) Comparison with other dictionary learning methods (i.e. not SAE) that are known to work well for extracting meaningful concepts (NMF, Sparse NMF...) would strengthen the comparison. This paper is oriented toward a specific task (i.e. concept extraction), so it would be great to compare the proposed algorithms with more concept extraction methods (and SAE is far from being the only good method to extract concepts).
4) As already mentioned before, quantifying the performance of SAEs based only on the reconstruction/sparsity tradeoff is not enough. It would be interesting to include additional metrics.


[1] Rajamanoharan, Senthooran, et al. "Jumping ahead: Improving reconstruction fidelity with jumprelu sparse autoencoders." arXiv preprint arXiv:2407.14435 (2024).

**Methods And Evaluation Criteria:**

1 ) Concerning the reconstruction/sparsity tradeoff comparison. The authors are comparing the different trade-off for topK and the ReLU SAE with the MSAE, by sampling only 3 points (i.e. different lambda of K). So few points seem to be not enough to make strong evidence for the first claim. I would urge the authors to rerun their comparison by systematically considering more lambda or K

2) Using only the reconstruction/sparsity tradeoff to compare various SAEs does not provide a complete view of the SAE performance. For example, articles have shown that SAE suffers from unstable concepts (i.e. 2 SAE trained with a different seed lead to different concepts). Such a property strongly affects the reproducibility of the SAE and impairs their use as a widespread interpretability tool (see [1]). So metrics quantifying the instability would also help to better evaluate the different SAE. But instability is just one example, there are plenty of interesting other metrics that would give a more complete comparison of the SAEs (see table 1, page 8 of [2], to have a non-exhaustive list of interesting metrics to evaluate SAEs).

[1] Paulo, Gonçalo, and Nora Belrose. "Sparse Autoencoders Trained on the Same Data Learn Different Features." arXiv preprint arXiv:2501.16615 (2025).
[2] Fel, Thomas, et al. "Archetypal SAE: Adaptive and Stable Dictionary Learning for Concept Extraction in Large Vision Models." arXiv preprint arXiv:2502.12892 (2025).

**Other Comments Or Suggestions:**

* It would be valuable to discuss or quantify the stability of concepts learned by MSAE, as stability significantly affects interpretability and practical utility in concept extraction tasks.
* Discuss more explicitly why the UW variant consistently outperforms RW on semantic metrics despite having lower sparsity, questioning the suitability of sparsity/reconstruction as the primary evaluation criterion.

**Other Strengths And Weaknesses:**

Strengths:
* Clear and well-justified approach.

Weaknesses:

* Limited comparative scope, omitting modern SAEs (e.g., JumpReLU) and alternative concept extraction methods.
* Lack of consideration for stability of SAE-derived concepts across training runs, a critical aspect for interpretability.

**Questions For Authors:**

* Could you include metrics or analyses to quantify the stability of the concepts learned by MSAE across different random initializations?

* Can you elaborate on why UW outperforms RW on all tested semantic preservation metrics, and discuss whether sparsity/reconstruction is indeed the best criterion for concept extraction performance?

* Why were alternative concept extraction methods (e.g., Convex-NMF, Sparse PCA, ICA) not included in your comparative analysis?

**Relation To Broader Scientific Literature:**

SAE approaches are well situated within existing literature; however, alternative concept extraction methods such as Convex-NMF, semi-NMF, Sparse PCA, ICA, SVD, and KMeans, known to perform well on similar tasks, are notably absent from comparisons.

**Theoretical Claims:**

There is no theoretical claims (this is an experimental article).

---

> ### Author Rebuttal · Authors · 2025-04-01
>
> **We gratefully thank the reviewer for a thorough and very insightful review of our work.** Due to space limits (5000 characters), we concisely respond to each of the major points raised. We share tables and figures with results from the requested analyses in anonymous cloud storage at https://drive.google.com/drive/folders/11OOSpexmU5ul8nBhJqpaf65Dh_eqBlZY
>
> A. **Reconstruction/sparsity tradeoff comparison.** (Mentioned in: Methods & Eval. 1, Exp. Designs 1)
> > The authors are comparing the different trade-off for topK and the ReLU SAE with the MSAE, by sampling only 3 points (i.e. different lambda of K). [...] I would urge the authors to rerun their comparison by systematically considering more lambda or K.
>
> Our initial experiments indeed span 4 different lambda and K hyperparameter values, as listed in Appendix B.1. We show the majority of results for 3 hyperparameter values due to observing low performance in the fourth one (lambda=0.01 & K=32, respectively). Following related work (Gao et al., 2024), we excluded the more extreme values of hyperparameters (e.g. K=512) since these lead to subpar results. As the reviewer suggested, **we now added the comparison for lambda=0.01 and K={32,512} to Tables 6 & 7** (see cloud link). We observe that lambda=0.01 gives worse results than MSAE, while K=512 collapses during training (FVU=0.3, CKNNA=0.07).
>
> B. **Stability of SAE-derived concepts across training runs.** (Methods & Eval. 2, Exp. Designs 4, Weakness 2, Other Comments 1, Question 1)
> > [...] Such a property strongly affects the reproducibility of the SAE and impairs their use as a widespread interpretability tool (see [1]). [...] But instability is just one example, there are plenty of interesting other metrics that would give a more complete comparison of the SAEs (see table 1, page 8 of [2], to have a non-exhaustive list of interesting metrics to evaluate SAEs). [1] "Sparse autoencoders trained on the same data learn different features." preprint arXiv:2501.16615 (2025). [2] "Archetypal SAE [...]" preprint arXiv:2502.12892 (2025). Could you include metrics or analyses to quantify the stability of the concepts learned by MSAE across different random initialization?
>
> Thank you for highlighting this work. **We politely note that neither of the mentioned papers was publicly available at the time of our paper’s submission.** The [2] preprint appeared over two weeks (18 Feb 2025) after the ICML submission deadline; similarly, the [1] preprint appeared a day before (29 Jan 2025). Not relating to literature that was unavailable at the time of submission should, in our opinion, neither be considered a weakness of our work nor affect the review score. Following the reviewer’s recommendation, **we now included an analysis regarding the stability of SAEs [1] for CLIP in the new Table 18** (see cloud link). We envision that applying the archetypal framework [2] to our proposed architecture would improve the stability of MSAE.
>
> C. **Comparison with JumpReLU.** (Exp. Designs 2, Weakness 1)
> > It would have been interesting to compare the MSAE to the jump ReLU SAE [1], as it is known to have a good reconstruction/sparsity tradeoff. [1] "Jumping ahead: Improving reconstruction fidelity with JumpReLU sparse autoencoders." preprint arXiv:2407.14435 (2024).
>
> **Note that the JumpReLU preprint has no official code implementation available.** During our initial work, we were unable to reproduce its results. We now used the unofficial implementation shared in the SAELens software to compare with MSAE (ImageNet-1k, CLIP ViT-L/14). In our preliminary experiments, the results of JumpReLU are the same as those of ReLU. Note that a similar issue with reproducibility has been publicly pointed out in “Interpretability evals case study” (Transformer Circuits Thread, Aug 2024). **Our result is visible in the updated Tables 6 & 7** (see cloud link).
>
> D. **Can you elaborate on why UW outperforms RW on all tested semantic preservation metrics, and discuss whether sparsity/reconstruction is indeed the best criterion for concept extraction performance?** (Comment 2, Question 2)
>
> In brief, RW is trained with an alpha parameter that values sparser representations more highly during training than UW, where alpha does not affect the loss. RW models learn to benefit from a sparser reconstruction, sacrificing optimal reconstruction in favor of sparsity, as indicated by lower FVU in UW. In our experiments, CKNNA and LP semantic metrics are correlated with reconstruction, while the quantity of valid concept neurons (Table 3) is correlated with sparsity.
>
> E. **Why were alternative concept extraction methods (e.g., Convex-NMF) not included in your comparative analysis?** (Exp. Designs 3, Q. 3)
>
> Thank you for bringing this work to our attention; we view such a comparison as a natural future work direction. We primarily focused on evaluating various SAE architectures for interpreting CLIP using multiple quantitative metrics; beyond LLM applications.

---

> > ### Comment · Reviewer_T2kP · 2025-04-08
> >
> > I agree that the listed were not released at the time of the submission, but these papers are just examples (among others that are older) to show the authors that plenty of other metrics could be used to meaningfully compare their SAE to other (stability is just one of them). I will keep rating, because I still think comparing different SAE just based on reconstruction/sparsity is not enough.

---

> > > ### Author Response · Authors · 2025-04-09
> > >
> > > We go beyond reconstruction-sparsity evaluation protocols from related work (JumpReLU, TopK), incorporating additional meaningful metrics: **CKNNA** alignment (Table 1) measuring how SAE activations align with original CLIP representations and **valid concept count** (Table 3) that quantities interpretability.
> > >
> > > Our results show that valid concept count correlates with sparsity, indicating sparsity signals interpretability. CKNNA reveals unique insights about SAE activations, e.g. in Figures 16-17 that only Matryoshka transfers both coarse and fine-grained features across domains, especially at expansion rate 32. Other methods' fine-grained SAE activations appear domain-specific or are used only for reconstruction.

---

### Official Review · Reviewer_26gq · 2025-03-14

**Overall Recommendation:** 3

**Summary:**

Sparse Autoencoders (SAEs) have been adopted to interpret CLIP’s feature representations, but the trade-off between reconstruction quality and sparsity has made it difficult to strike an ideal balance for interpretation. This paper proposes Matryoshka Sparse Autoencoder (MSAE), a hierarchical extension of SAEs, to analyze CLIP’s representation space in multiple granularities. By doing so, MSAE can extract more than 120 semantic concepts from CLIP’s embedding space and demonstrate its applicability in concept-based similarity matching (with controllable concept strength) and gender bias analysis for downstream tasks.

## Update after rebuttal
I agree with the concern raised by other reviewers regarding the lack of quantitative comparisons. The authors' response does not seem to adequately address this issue. Therefore, I have lowered my score.

**Claims And Evidence:**

**Claim 1**: Existing SAE-based methods face limitations when interpreting CLIP’s multimodal representations.
- Evidence: Prior studies indicate that simple L1 regularization can undervalue important features, while a strict TopK approach enforces overly rigid sparsity, imposing interpretability constraints.

-> The authors cite some earlier works convincingly, making a clear and logical argument.

**Claim 2**: MSAE achieves a superior balance between sparsity and reconstruction quality compared to existing SAE baselines.
- Evidence: As the shown in Figure 2, MSAE consistently outperforms standard ReLU and TopK SAEs on EVR vs. L0.

-> Experimental results show clear and convincing evidence.

**Claim 3**: MSAE facilitates more robust interpretation of CLIP’s multimodal representation.
-  Evidence: By applying MSAE to CLIP’s embedding space, the authors discover more than 120 interpretable concepts (Table 3).

-> Experimental results show clear and convincing evidence.

**Essential References Not Discussed:**

There appear to be no critical references missing from the paper's discussion.

**Experimental Designs Or Analyses:**

The authors highlight the original issue—finding a workable sparsity-fidelity trade-off—and thoroughly address it using multiple metrics. Their design effectively analyzes how MSAE balances reconstruction precision against high sparsity. Moreover, ablation studies illustrate the hierarchical advantage of MSAE over single-threshold SAEs.

**Methods And Evaluation Criteria:**

- The proposed multi-TopK approach is well aligned with the goal of preserving hierarchical structure in CLIP embeddings.
- The evaluation metrics effectively measure reconstruction fidelity.

**Other Comments Or Suggestions:**

It might help to clarify in Section 5 exactly where and why alternative SAEs (e.g., single-threshold TopK) fall short for concept-based similarity or bias detection. This would further underscore MSAE’s value for real interpretability applications.

**Other Strengths And Weaknesses:**

**Strengths**:
- The paper systematically evaluates MSAE under multiple metrics, convincingly showing that MSAE attains a better sparsity-fidelity trade-off than existing approaches.
- Through experiments in Section 5, the authors illustrate how MSAE enables meaningful interpretation of CLIP, including concept-based similarity matching and downstream bias analysis, demonstrating its real-world applicability.

**Weaknesses**:
- While Section 4 demonstrates the advantages of MSAE over standard SAEs, Section 5 primarily focuses on MSAE’s applications—showing concept extraction and bias analysis with CLIP. It would have been even stronger if the paper made explicit how previous SAEs would fail or underperform in these same tasks.

**Questions For Authors:**

* In Section 5.3, the authors train a single-layer classifier on CLIP embeddings to identify bias for the CelebA experiment. Couldn’t the text encoder in CLIP itself be used for classification without additional training? Why did you decide to train a new classifier for this bias detection step?

**Relation To Broader Scientific Literature:**

This work stands at the intersection of mechanistic interpretability and concept-based explanations for large vision-language models. It also addresses limitations in prior SAEs, bridging techniques like L1-based ReLU and fixed TopK with a flexible hierarchical approach.

**Theoretical Claims:**

There is no theoretical claim.

---

> ### Author Rebuttal · Authors · 2025-04-01
>
> **We sincerely thank the reviewer for acknowledging the quality and significance of our work.**
>
> > While Section 4 demonstrates the advantages of MSAE over standard SAEs, Section 5 primarily focuses on MSAE’s applications—showing concept extraction and bias analysis with CLIP. It would have been even stronger if the paper made explicit how previous SAEs would fail or underperform in these same tasks. [...] It might help to clarify in Section 5 exactly where and why alternative SAEs (e.g., single-threshold TopK) fall short for concept-based similarity or bias detection. This would further underscore MSAE’s value for real interpretability applications.
>
> Thank you for raising this point. We agree with the reviewer that a measurement of interpretability would further strengthen our paper and view it as a natural future work direction. To the best of our knowledge, **our work is the first to evaluate well-established SAE architectures using multiple quantitative metrics beyond LLM applications**. Given this scope, there is a limit to how many new contributions a single paper can explore exhaustively (new architecture, bi-modal setting, state-of-the-art performance, potential applications, and also new evaluation protocol).
>
> To facilitate research into this direction, we supplement our work with additional visual examples akin to Figure 10. **We now share figures with explanations of 502 concepts across 6 different SAEs (appearing in Table 2) in anonymous cloud storage at** https://drive.google.com/drive/folders/11OOSpexmU5ul8nBhJqpaf65Dh_eqBlZY  (files `concept_visualization_*.pdf`). Each figure presents inputs the most activating a given concept across both modalities (vision and language). For a broad overview, we visualize all validated concepts (see a discussion in Appendix A) appearing in any of these 6 models (see the last column in Table 3).
>
> > In Section 5.3, the authors train a single-layer classifier on CLIP embeddings to identify bias for the CelebA experiment. Couldn’t the text encoder in CLIP itself be used for classification without additional training? Why did you decide to train a new classifier for this bias detection step?
>
> Yes, the CLIP's text encoder can generally be used for zero-shot classification. However, we chose to fine-tune a classifier on top of CLIP to demonstrate the broader applicability of MSAE to interpret other models. For example, another classifier trained on the CLIP’s representation, or even another feature extractor (backbone) model that lacks zero-shot classification capabilities, unlike CLIP. We use MSAE to generate counterfactual explanations, which are particularly valuable in bias-sensitive predictive tasks like CelebA.

---

### Official Review · Reviewer_Zxzv · 2025-03-17

**Overall Recommendation:** 3

**Summary:**

This article explains the CLIP model from the perspective of model parameters. The author uses sparse autoencoders to sparse the content learned by the neurons of the CLIP model. Specifically, the author proposes Matryoshka Sparse Autoencoder, which is a hierarchical encoder used to hierarchicalize conceptual information when training SAE. The author has done a lot of applications to prove the practicality of this method.

## update after rebuttal

**Claims And Evidence:**

The author mentioned in the introduction that the previous SAE method was limited by Top-K and L1 losses, and the hierarchical learning proposed in this paper can overcome these limitations. Although the key issues are mentioned, it seems unclear how the method in this paper solves these problems.

**Essential References Not Discussed:**

Not sure

**Ethical Review Flag:**

Flag this paper for an ethics review.

**Experimental Designs Or Analyses:**

The authors designed many downstream applications to demonstrate the practicality of the proposed method.

**Methods And Evaluation Criteria:**

The authors adopted the evaluation metrics commonly used in previous work to evaluate SAE.

**Other Comments Or Suggestions:**

NA

**Other Strengths And Weaknesses:**

**Strengths:** It is very meaningful to study the use of sparse autoencoders to explain model parameters. The author proposed hierarchical learning to distribute the concept learning of sparse autoencoders from coarse-grained to fine-grained, and demonstrated the practicability of the method in multiple downstream tasks.

**Weaknesses:**

- The authors seem to lack quantitative comparisons with some existing SAE-based neuron interpretability methods, only some simple ReLu and TopK-based methods.

- Similarly, although the method is practical in downstream tasks, it is difficult to intuitively feel the degree to which it surpasses advanced methods. I suggest that the author consider supplementing the latest comparison method, and then try to explain the same parameters of the same model, and then conduct some applications to help us intuitively feel the difference in the ability of different SAE-based methods to explain parameters.

- The author explains CLIP throughout the paper, but the method in this paper does not seem to be a method for explaining the CLIP model specifically, and it can still be applied to other single-modal models. The author needs to clarify the relationship between the method in this paper and CLIP. Is it a specific design, or can it be universal? (If it is universal, it is recommended that the author can add some experiments for interpreting other model parameters).

**Questions For Authors:**

NA

**Relation To Broader Scientific Literature:**

The research in this article helps promote the development of explainable AI, especially parameter-level explanations.

**Theoretical Claims:**

No theoretical analysis in the article, if I miss it, please remind me.

---

> ### Author Rebuttal · Authors · 2025-04-01
>
> **We gratefully thank the reviewer for their engagement with our work and appreciation of our contribution.**
>
> > The authors seem to lack quantitative comparisons with some existing SAE-based neuron interpretability methods, only some simple ReLu and TopK-based methods.
>
> Existing SAE-based methods have been primarily compared in interpreting language models, **while our work goes beyond to specifically interpret vision–language models instead**. Therefore, we choose ReLU and TopK as the potential baseline methods. As suggested by Reviewer #T2kP, we now attempted to compare with JumpReLU SAE. Unfortunately, the JumpReLU preprint has no official code implementation available. We now used the unofficial implementation shared in the SAELens software to compare with MSAE (ImageNet-1k, CLIP ViT-L/14). In our preliminary experiments, the results of JumpReLU are virtually the same as those of ReLU. Note that a similar issue with reproducibility has been publicly pointed out in “Interpretability evals case study” (Transformer Circuits Thread, August 2024). We now share our result in the updated Tables 6 & 7 in anonymous cloud storage at https://drive.google.com/drive/folders/11OOSpexmU5ul8nBhJqpaf65Dh_eqBlZY (file `a_updated_tables_6_and_7.pdf`).
>
> We initially also considered comparing with Gated SAE, but acknowledged that TopK outperformed it based on the results from (Gao et al., 2024).
>
> > Similarly, although the method is practical in downstream tasks, it is difficult to intuitively feel the degree to which it surpasses advanced methods. I suggest that the author consider supplementing the latest comparison method, and then try to explain the same parameters of the same model, and then conduct some applications to help us intuitively feel the difference in the ability of different SAE-based methods to explain parameters.
>
> Thank you for highlighting the practical utility of our method. Kindly, it is unclear to us which “advanced methods” were meant by the reviewer. We agree it is challenging to provide an intuitive feeling regarding how different SAE architectures explain the model. Thus, **we rely on 5 quantitative metrics established in the literature, and introduce 2 novel metrics (CKNNA and DO) to assess MSAE against baselines**, an evaluation previously unexplored for CLIP (Section 4). We then demonstrate the best-performing SAE's utility in three downstream tasks (Section 5), providing visual explanations to further illustrate its effectiveness (Appendix Figures 18–23).
>
> To address the reviewer’s suggestion, we supplement our work with additional visual examples akin to Figure 10. **We now share figures with explanations of 502 concepts across 6 different SAEs (appearing in Table 2) in anonymous cloud storage at** https://drive.google.com/drive/folders/11OOSpexmU5ul8nBhJqpaf65Dh_eqBlZY (files `concept_visualization_*.pdf`) to facilitate building up this intuitive feeling. Each figure presents inputs the most activating a given concept across both modalities (vision and language). For a broad overview, we visualize all validated concepts (see a discussion in Appendix A) appearing in any of these 6 models (see the last column in Table 3).
>
> > The author explains CLIP throughout the paper, but the method in this paper does not seem to be a method for explaining the CLIP model specifically, and it can still be applied to other single-modal models. The author needs to clarify the relationship between the method in this paper and CLIP. Is it a specific design, or can it be universal? (If it is universal, it is recommended that the author can add some experiments for interpreting other model parameters).
>
> This is a good point. While we demonstrate our method on the multi-modal CLIP model for broader appeal (encompassing both vision and language), our approach is indeed applicable to single-modal models as well. We chose CLIP due to its widespread interest and the potential for insightful interpretations across modalities. Unlike much of the previous work, which focused on language models, we apply SAE to interpret vision–language models often used as feature extractors. Application of SAE to the final representation of the feature extractor like CLIP avoids the issue of concept re-emergence after steering, which remains a challenge in LLM applications. Note that our steering presented in Sections 5.2 & 5.3 uses the CLIP’s final representations, circumventing this problem.
>
> We appreciate the reviewer’s feedback and will incorporate the discussion in the next version of the paper.

---

> > ### Comment · Reviewer_Zxzv · 2025-04-07
> >
> > I appreciate the author's response, and after final consideration, I have decided to keep my score.

---

### Decision · Program_Chairs · 2025-05-01

**Decision:**

Accept (poster)

**Comment:**

This paper introduces Matryoshka Sparse Autoencoder (MSAE), a novel architecture that addresses the fundamental trade-off between reconstruction quality and sparsity in sparse autoencoders when interpreting vision-language models like CLIP. By learning hierarchical representations at multiple levels of granularity simultaneously, MSAE achieves superior performance compared to traditional approaches such as ReLU and TopK sparse autoencoders. The reviewers acknowledged several strengths of the paper, including the clear methodology and strong empirical results showing MSAE's ability to establish a new state-of-the-art Pareto frontier between reconstruction quality and sparsity for CLIP. The practical applications demonstrating MSAE's utility as an interpretability tool for extracting semantic concepts and performing concept-based similarity search and bias analysis were also highlighted positively. Some concerns were raised regarding limited comparative analysis with more recent SAE variants and alternative concept extraction methods, as well as insufficient evaluation metrics beyond reconstruction/sparsity trade-offs. The authors addressed these concerns in their rebuttal by conducting additional experiments comparing with JumpReLU SAE, adding stability analysis, and clarifying their contribution of two new evaluation metrics. Given the novel approach to interpreting multimodal representations, the comprehensive experiments, and the demonstrated practical applications for understanding CLIP, I recommend accepting this paper for publication at ICML 2025.